# DISTILLKAC: FEW-STEP IMAGE GENERATION VIA DAMPED WAVE EQUATIONS

**Weiqiao Han**
MIT

**Chenlin Meng**
Stanford

**Christopher D. Manning**
Stanford

**Stefano Ermon**
Stanford

## ABSTRACT

We present DistillKac, a fast image generator that uses the damped wave equation and its stochastic Kac representation to move probability mass at finite speed. In contrast to diffusion models whose reverse time velocities can become stiff and implicitly allow unbounded propagation speed, Kac dynamics enforce finite speed transport and yield globally bounded kinetic energy. Building on this structure, we introduce classifier-free guidance in velocity space that preserves square integrability under mild conditions. We then propose endpoint only distillation that trains a student to match a frozen teacher over long intervals. We prove a stability result that promotes supervision at the endpoints to closeness along the entire path. Experiments demonstrate DistillKac delivers high quality samples with very few function evaluations while retaining the numerical stability benefits of finite speed probability flows.

## 1 INTRODUCTION

Diffusion models have catalyzed an enormous research wave. Since the seminal Denoising Diffusion Probabilistic Models (DDPM) paper, there are now tens of thousands of diffusion-related publications, and the DDPM work alone has accrued well over twenty-five thousand citations (Ho et al., 2020; Sohl-Dickstein et al., 2015). Conceptually, diffusion models couple a stochastic forward noising process with a reverse-time ordinary differential equation (ODE)/stochastic differential equation (SDE) whose density evolution is governed by the Fokker–Planck equation, a linear second-order partial differential equation (PDE) (Song et al., 2021b). Learning consists of estimating time-indexed scores/velocities and integrating them for sampling (Ho et al., 2020; Song & Ermon, 2020; Song et al., 2021a) Recently, a growing body of work explores probability flow backbones: deterministic ODE flows trained via flow matching and its optimal transport variants (Lipman et al., 2023; Tong et al., 2024), rectified flows (Liu et al., 2023), and unified stochastic interpolants that bridge flows and diffusion (Albergo et al., 2023).

Yet the PDE lens on generative modeling need not be limited to Fokker–Planck. Other classical PDEs offer different structural advantages but remain comparatively underexplored. For example, Poisson flow models recast generation through electrostatics-inspired fields (Xu et al., 2022; 2023). Most relevant to our work, Duong et al. (2026) replaces the Fokker–Planck equation with the telegrapher equation (1-dimensional damped wave equation) and its stochastic Kac representation, proving finite-speed probability flows that are Lipschitz in Wasserstein distance with globally bounded velocity norms. Crucially, telegrapher/Kac dynamics enforce a finite propagation speed: starting from localized mass, the distribution remains inside a causal cone. In contrast, diffusion spreads mass instantaneously with effectively infinite propagation speed, which also manifests as stiff, rapidly growing reverse-time velocities near terminal time (Yang et al., 2023). Indeed, diffusion arises as a limit of the damped wave equation as the damping and propagation speed tend to infinity. This structural gap, finite speed versus infinite-speed propagation, will be central to our modeling and algorithms.

In this paper, we pivot to the damped wave equation and its stochastic Kac representation, which induces finite-velocity probability flows. Compared to diffusion ODEs/SDEs, whose velocity norms can blow up near terminal time, Kac dynamics yield globally bounded kinetic energy and Lipschitz regularity in Wasserstein space. Intuitively, the finite-speed cap $c$ acts like a built-in stability constraint: trajectories cannot accelerate without bound, characteristic fronts move no faster than $c$, and

| Parabolic PDE | Elliptic PDE | Hyperbolic PDE |
|---|---|---|
| Fokker–Planck equation | Poisson equation | Damped wave equation |
| Diffusion models | Poisson flow models | Kac flow models |
| (Sohl-Dickstein et al., 2015; Ho et al., 2020) and tens of thousands of other diffusion papers | (Xu et al., 2022; 2023) | (Duong et al., 2026) and this work |

Table 1: Three PDE lenses for generative modeling: representative model families and citations. Our work belongs to the hyperbolic family, shown in the third column.

late-time integration is less stiff. Building on this structure, we introduce guided Kac flows for conditional generation and distilled Kac flows for few-step sampling: (i) we adapt classifier-free guidance (Ho & Salimans, 2022) directly in velocity space while preserving bounded-energy guarantees, and (ii) we develop an endpoint-only distillation scheme with a provable endpoint-to-trajectory stability bound. Empirically, these properties support stable few-step samplers with strong sample quality. Theoretically, they clarify why endpoint matching suffices under finite-speed Kac dynamics.

## 2 THEORETICAL FOUNDATIONS

### 2.1 PARTIAL DIFFERENTIAL EQUATIONS

The time evolution of the probability density of the diffusion process is governed by the *Fokker-Planck equation*

$$\partial_t p(t, x) = -\nabla \cdot (p\mu)(t, x) + \sum_{i,j} \partial_{x_i} \partial_{x_j} (D_{ij}(t, x)p(t, x)) \tag{1}$$

where $t \in \mathbb{R}$ is time, $x \in \mathbb{R}^d$ is state, $\mu(t, x)$ is the drift vector, and $D(t, x) = \frac{1}{2}\sigma\sigma^\top$ is the diffusion tensor. The *heat equation* is a special case of the Fokker-Planck equation.

While the Fokker-Planck equation is a linear, second-order parabolic PDE, there are other important classes of linear second-order PDEs, such as elliptic and hyperbolic equations. In particular, the wave equation and its damped variant are canonical examples of hyperbolic PDEs. The *damped wave equation* takes the general form

$$\partial_{tt} u(t, x) + \xi \partial_t u(t, x) = c^2 \Delta u(t, x) \tag{2}$$

where $\xi > 0$ is a constant and $c$ is the speed of the wave front. The *telegraph(er) equation* is a 1-dimensional damped wave equation.

### 2.2 KAC PROCESS AND RANDOM FLIGHTS

Just as the diffusion process is associated with the Fokker-Planck equation, the damped wave equation also corresponds to a stochastic process.

Define a stochastic process $\{X(t)\}_{t \geq 0}$ in $\mathbb{R}^d$, $d \geq 1$, as follows:

1. Initial condition. The particle starts at the origin $X(0) = 0 \in \mathbb{R}^d$.
2. Velocity and speed. The particle moves with constant speed $c > 0$. At any time, the velocity is $V(t) = cU(t)$, where $U(t)$ is a random vector on the unit sphere $\mathbb{S}^{d-1} = \{u \in \mathbb{R}^d : \|u\| = 1\}$.
3. Direction changes. The direction process $\{U(t)\}$ changes at the jump times of a homogeneous Poisson process $\{N(t)\}_{t \geq 0}$ of rate $\lambda > 0$. That is, if the jump times are $0 = s_0 < s_1 < s_2 < \cdots < s_n$, then:
   - On each interval $[s_{j-1}, s_j)$, the velocity is constant: $V(t) = cU_j$.
   - The random directions $\{U_j\}$ are i.i.d. with uniform distribution on the unit sphere $\mathbb{S}^{d-1}$.

The position of the particle at time $t$ is obtained by integrating the velocity

$$X(t) = \int_0^t V(s)ds = c \sum_{j=1}^{N(t)} (s_j - s_{j-1})U_j, \qquad (3)$$

where $s_0 = 0$ and $s_{N(t)} = t$.

When $d = 1$, $\mathbb{S}^0 = \{\pm 1\}$, and the particle flips directions between left and right. The $d = 1$ case is usually called *telegraph process* or *Kac process* (Goldstein, 1951; Kac, 1974). When $d \geq 2$, the process is called a *random flight* (Orsingher & De Gregorio, 2007).

**Theorem 1.** *(Orsingher & De Gregorio, 2007) For $d = 1, 2$, and $4$, the distribution of the state $X(t)$ can be obtained analytically, while for any other positive integer $d$ there is no explicit formula.*

**Theorem 2.** *(Orsingher & De Gregorio, 2007) For $d = 1$ and $2$, the state distribution solves the damped wave equation.*

## 2.3 IMAGE GENERATION VIA COMPONENT-WISE 1-D TELEGRAPHER EQUATION

Duong et al. (2026) study Kac process and the telegrapher equation (the 1-D damped wave equation), and derive a closed-form expression for the conditional velocity $v(t, x|x_0)$ for any time $t$, and noisy state $x \in \mathbb{R}$ given data $x_0$. Leveraging this expression, they propose a simple procedure to learn the data distribution from noise:

1. Given data $x_0$, sample a time $t$ and then draw the state $x$ from the analytic state distribution (Theorem 1).

2. Evaluate the closed-form conditional velocity $v(t, x|x_0)$ and fit a neural network $v_\theta(t, x)$ to it by regression (e.g. minimizing $\mathbb{E}[\|v_\theta(t, x) - v(t, x|x_0)\|^2]$).

At inference, draw $x_T \sim p(T, x)$ (noise) and integrate the learned velocity field backward in time from $t = T$ to $t = 0$ to obtain a data sample $x_0$.

For multi-dimensional data, e.g., images, they model each coordinate with an independent 1-D Kac process, i.e., a component-wise product construction.

Because each coordinate follows the 1-D Kac dynamics, the component-wise image model inherits finite-speed propagation and dimension-aware energy bounds. In 1-D, the Kac solution started at $x_0$ is supported on $[x_0 - ct, x_0 + ct]$, where $c$ is the wave front speed in Equation (2), and the conditional velocity satisfies $|v(t, x|x_0)| \leq c$, with $v(t, \pm ct|x_0) = \pm c$. Applying this per coordinate, the component-wise construction for images propagates inside the $\ell_\infty$ causal cone $\{x : \|x - x_0\|_\infty \leq ct\}$. Moreover, for the resulting $d$-dimensional (product) Kac flow $(\mu_t)_{t \geq 0}$ there exist dimension-aware global bounds

$$W_2(\mu_s, \mu_t) \leq c\sqrt{d}|t - s|, \qquad \|v_t\|_{L^2(\mu_t)} \leq c\sqrt{d} \quad \text{for a.e. } t \in [0, 1], \qquad (4)$$

and the support grows at most linearly in time. We will leverage these properties to prove bounded energy under classifier-free guidance and endpoint-to-trajectory stability for distillation.

## 2.4 NOTATIONS

Let $t \in \mathbb{R}$ denote time, $x \in \mathbb{R}^d$ the state, and $y \in \mathbb{N}$ a class label. Let $\mathcal{P}_2(\mathbb{R}^d)$ denote the set of Borel probability measures on $\mathbb{R}^d$ with finite second moment, endowed with the quadratic Wasserstein distance

$$W_2^2(\mu, \nu) := \inf_{\pi \in \Pi(\mu, \nu)} \int_{\mathbb{R}^d \times \mathbb{R}^d} \|x - y\|^2 \mathrm{d}\pi(x, y), \qquad (5)$$

for $\mu, \nu \in \mathcal{P}_2(\mathbb{R}^d)$, where $\Pi(\mu, \nu)$ is the set of couplings of $\mu$ and $\nu$. For a measurable map $F : \mathbb{R}^d \to \mathbb{R}^d$, the pushforward measure of $\mu$ is $F_\# \mu := \mu \circ F^{-1}$.

We use *Kac process* when we talk about individual random paths (simulation/noising), and *Kac flow* when we talk about how distributions move in time and the ODE we integrate at inference.

---

**Algorithm 1** Endpoint Distillation ($N$ steps)

---

1: **Inputs:** $N$, $\Delta t$, $\tilde{v}_\theta$ (normal/CFG velocity), $\eta$, $B$, $max\_iter$
2: **Init:** teacher $u_\theta \leftarrow \tilde{v}_\theta$ (frozen), student $v_\theta \leftarrow \tilde{v}_\theta$
3: **for** $i = 1$ **to** $max\_iter$ **do**
4:      sample $\{(t^{(b)}, x^{(b)}, y^{(b)})\}_{b=1}^B$
5:      $x_\star^{(b)} \leftarrow \text{TEACHER}(u_\theta, t^{(b)}, x^{(b)}, y^{(b)}, N, \Delta t)$
6:      $\hat{x}^{(b)} \leftarrow x^{(b)} - v_\theta(t^{(b)}, x^{(b)}; y^{(b)})\Delta t$
7:      $\mathcal{L}(\theta) \leftarrow \frac{1}{B}\sum_b \|\hat{x}^{(b)} - x_\star^{(b)}\|_2^2; \quad \theta \leftarrow \theta - \eta\nabla_\theta\mathcal{L}(\theta).$
8: **end for**

---

## 3   CLASSIFIER FREE GUIDANCE IN VELOCITY SPACE

While Duong et al. (2026) focus on unconditional image generation, we address both unconditional and conditional generation, and further introduce classifier-free guidance in velocity space.

Let $v_\theta(t, x)$ and $v_\theta(t, x; y)$ be the learned unconditional and conditional velocities, respectively. We define the velocity guidance with strength $w(t) \geq 0$ by

$$\tilde{v}(t, x; y) = v_\theta(t, x) + w(t)[v_\theta(t, x; y) - v_\theta(t, x)], \tag{6}$$

and integrate the reverse ODE $\dot{\varphi}(t) = -\tilde{v}(t, \varphi(t); y)$ from $t = 1$ to $0$ to generate data samples.

Let $\Delta_\theta(t, x; y) := v_\theta(t, x; y) - v_\theta(t, x)$ denote the conditional-unconditional velocity gap.

**Assumption 3.** *(Square-integrable guidance gap) For a.e. $t \in [0, 1]$, $\mathbb{E}_{x\sim\mu_t}[\|\Delta_\theta(t, x; y)\|_2^2] < \infty$ for the random condition $y$, or equivalently, $\mathbb{E}_y\mathbb{E}_{x\sim\mu_t}[\|\Delta_\theta(t, x; y)\|_2^2] < \infty.$*

**Theorem 4.** *(Energy bound under guidance) Suppose the unconditional Kac flow satisfies $\|v_\theta(t, \cdot)\|_{L^2(\mu_t)} \leq c\sqrt{d}$ for a.e. $t$, and Assumption 3 holds. If $|w(t)|$ is finite for a.e. $t$, then $\tilde{v}(t, \cdot; y) \in L^2(\mu_t)$ for a.e. $t$.*

The proof is in the Appendix. The assumption that $\|v_\theta(t, \cdot)\|_{L^2(\mu_t)} \leq c\sqrt{d}$ for a.e. $t$ follows from a corresponding $L^2(\mu_t)$ bound on the velocity field in Duong et al. (2026) under its stated hypothesis, which is generally satisfied. In practice, we may take $w(t)$ to be a constant $w$, so the requirement that $|w(t)|$ is finite for a.e. $t$ is automatically satisfied. The conclusion $\tilde{v}(t, \cdot; y) \in L^2(\mu_t)$ for a.e. $t$ then implies that the guided velocity has finite kinetic energy, unlike diffusion, where the kinetic energy can diverge near data time.

## 4   DISTILLATION

We distill a student from a frozen teacher by endpoint matching. The teacher is either a learned flow or a classifier-free guidance model. Unlike progressive distillation (Salimans & Ho, 2022), which performs two-step distillation per iteration, we match over an arbitrary number of $N$ substeps with $N \geq 2$. The distillation algorithm is shown in Algorithm 1. At each training iteration and for each sample $b$, the TEACHER routine integrates the frozen teacher velocity $u_\theta$ backward from $t^{(b)}$ to $t^{(b)} - \Delta t$ using $N$ uniform substeps (e.g., explicit Euler, midpoint/RK2, Adams-Bashforth-2), producing the reference endpoint $x_\star^{(b)}$. The endpoints $t^{(b)}$ and $t^{(b)} - \Delta t$ are chosen to coincide with the student's segment boundaries: for an $M$-step student with a uniform time grid $1 = t_1 \geq \cdots \geq t_{M+1} = 0$, if $t^{(b)} = t_k$, then $t^{(b)} - \Delta t = t_{k+1}$. Over the same interval $[t^{(b)} - \Delta t, t^{(b)}]$, the student takes one explicit Euler step (backward in time), and is trained with an MSE loss to match $x_\star^{(b)}$. Both the teacher and the student are initialized from the same pre-trained velocity model $\tilde{v}_\theta$, and the teacher remains frozen throughout training.

Let $u$ and $v$ be the velocity fields for the teacher and the student Kac flows, respectively. The corresponding ODEs defined on $[0, 1]$ are

$$\dot{\varphi}_t = u(t, \varphi_t; y) \tag{7}$$

$$\dot{\psi}_t = v(t, \psi_t; y) \tag{8}$$

Let $\mu_t$ and $\nu_t$ be the respective distributions of $\varphi_t$ and $\psi_t$. Let $\Phi_{s\to t}$ and $\Psi_{s\to t}$ denote the flows associated with $\varphi_t$ and $\psi_t$, respectively. That is, for $0 \le s \le t \le 1$, the flow map $\Phi_{s\to t} : \mathbb{R}^d \to \mathbb{R}^d$ is the unique map such that for every $x \in \mathbb{R}^d$, the trajectory $\tau \mapsto \Phi_{s\to\tau}(x)$ is absolutely continuous on $[s, t]$ and solves the ODE Equation (7) for a.e. $\tau \in [s, t]$, with the initial condition $\varphi_s = x$:

$$\Phi_{s\to t}(x) = x + \int_s^t u(r, \Phi_{s\to r}(x); y)\mathrm{d}r. \tag{9}$$

The flow $\Phi_{s\to t}$ is also well defined when $0 \le t \le s \le 1$. In particular, sampling from noise $(t = 1)$ to data $(t = 0)$ corresponds to the single map $\Phi_{1\to 0}$.

We now lay out mild conditions on the teacher and student dynamics that ensure the flows are well defined. The subsequent assumptions and lemmas formalize these regularity and boundedness requirements and show that small discrepancies do not explode as we evolve in time. With this groundwork in place, the main stability result will convert endpoint agreement into closeness of entire trajectories, and the concluding corollary will turn that principle into concrete accuracy guarantees for practical, few-step students.

**Assumption 5.** *(Spatial Lipschitz drifts) There exists $L(t)$ such that, for a.e. $t \in [0, 1]$, both $u(t, \cdot; y)$ and $v(t, \cdot; y)$ are $L(t)$-Lipschitz in $x$ on the support of $\mu_t \cup \nu_t$, and $\int_0^1 L(t)\mathrm{d}t < \infty$.*

**Lemma 6.** *(Lipschitz constants of flow maps) Under Assumption 5, for any $0 \le s \le t \le 1$, $Lip(\Phi_{s\to t}) \le \exp(\int_s^t L(r)\mathrm{d}r)$, and similarly for $\Psi_{s\to t}$.*

The proofs of this lemma and the next are provided in the appendix. Here $Lip(\Phi_{s\to t})$ denotes the Lipschitz constant of the flow $\Phi_{s\to t}$.

**Lemma 7.** *(Pushforward contraction and coupling) For any $Lip(F)$-Lipschitz map $F : \mathbb{R}^d \to \mathbb{R}^d$ and $\mu, \nu \in \mathcal{P}_2(\mathbb{R}^d)$, $W_2(F_\#\mu, F_\#\nu) \le Lip(F)W_2(\mu, \nu)$. Moreover, for any probability measure $\nu \in \mathcal{P}_2(\mathbb{R}^d)$, and measurable maps $F, G : \mathbb{R}^d \to \mathbb{R}^d$, with $\int \|F(x)\|^2\mathrm{d}\nu(x) < \infty$ and $\int \|G(x)\|^2\mathrm{d}\nu(x) < \infty$, one has $W_2(F_\#\nu, G_\#\nu) \le (\mathbb{E}_{X\sim\nu}\|F(X) - G(X)\|_2^2)^{1/2}$.*

The following result captures the core stability principle behind endpoint distillation: if a student matches the teacher at the end of each interval, then the two paths remain close throughout that interval.

**Theorem 8.** *(Endpoint-to-trajectory stability) Let $\varepsilon := W_2(\mu_s, \nu_s)$ be the discrepancy at some time $s \in [0, 1]$. Under Assumption 5, for every $\tau \in [s, 1]$,*

$$W_2(\mu_\tau, \nu_\tau) \le \exp\left(\int_s^\tau L(r)\mathrm{d}r\right) \varepsilon + \int_s^\tau \exp\left(\int_t^\tau L(r)\mathrm{d}r\right) \|u(t, \cdot; y) - v(t, \cdot; y)\|_{L^2(\nu_t)}\mathrm{d}t \tag{10}$$

*Proof.* By well-posedness, $\mu_\tau = \Phi_{s\to\tau}{}_\#\mu_s$ and $\nu_\tau = \Psi_{s\to\tau}{}_\#\nu_s$. Apply the triangle inequality and Lemma 7,

$$W_2(\mu_\tau, \nu_\tau) \le W_2(\Phi_{s\to\tau}{}_\#\mu_s, \Phi_{s\to\tau}{}_\#\nu_s) + W_2(\Phi_{s\to\tau}{}_\#\nu_s, \Psi_{s\to\tau}{}_\#\nu_s) \tag{11}$$

$$\le Lip(\Phi_{s\to\tau})W_2(\mu_s, \nu_s) + (\mathbb{E}_{X\sim\nu_s}\|\Phi_{s\to\tau}(X) - \Psi_{s\to\tau}(X)\|_2^2)^{1/2} \tag{12}$$

By Lemma 6, $Lip(\phi_{s\to\tau}) \le \exp(\int_s^\tau L(r)\mathrm{d}r)$, giving the first term.

For the second term, define $\Delta_t(X) := \Phi_{s\to t}(X) - \Psi_{s\to t}(X)$ for $X \sim \nu_s$. Then $\frac{d}{dt}\Delta_t = u(t, \Phi_{s\to t}(X); y) - v(t, \Psi_{s\to t}(X); y)$. We have

$$\frac{d}{dt}\|\Delta_t\| \le \|\frac{d}{dt}\Delta_t\| \quad \text{(by Cauchy–Schwarz inequality, see Equation (A-8)} \tag{13}$$

$$= \|u(t, \Phi_{s\to t}(X); y) - v(t, \Psi_{s\to t}(X); y)\| \tag{14}$$

$$\le \underbrace{\|u(t, \Phi_{s\to t}(X); y) - u(t, \Psi_{s\to t}(X); y)\|}_{\le L(t)\|\Delta_t\| \text{ (by Assumption 5)}}$$

$$+ \underbrace{\|u(t, \Psi_{s\to t}(X); y) - v(t, \Psi_{s\to t}(X); y)\|}_{=:\delta(t, \Psi_{s\to t})} \quad \text{(by triangle inequality)} \tag{15}$$

By Grönwall's inequality with $\|\Delta_s\| = 0$,

$$\|\Delta_\tau(X)\| \le \int_s^\tau \exp\left(\int_t^\tau L(r)\mathrm{d}r\right) \delta(t, \Psi_{s\to t}(X))\mathrm{d}t. \tag{16}$$

Take $L^2(\nu_s)$ norms of both sides,

$$\|\Delta_\tau(X)\|_{L^2(\nu_s)} \le \left\|\int_s^\tau \exp\left(\int_t^\tau L(r)\mathrm{d}r\right) \delta(t, \Psi_{s\to t}(X))\mathrm{d}t\right\|_{L^2(\nu_s)} \tag{17}$$

$$\le \int_s^\tau \exp\left(\int_t^\tau L(r)\mathrm{d}r\right) \|\delta(t, \Psi_{s\to t}(X))\|_{L^2(\nu_s)}\mathrm{d}t \tag{18}$$

$$= \int_s^\tau \exp\left(\int_t^\tau L(r)\mathrm{d}r\right) \|u(t,\cdot;y) - v(t,\cdot;y)\|_{L^2(\nu_t)}\mathrm{d}t. \tag{19}$$

In (18) we used Minkowski's integral inequality. In (19), we used pushforward identity $\Psi_{s\to t\#}\nu_s = \nu_t$. $\qquad\square$

Setting $s = 0$, i.e., supervising only the data endpoint at $t = 0$, yields, for all $\tau \in [0, 1]$,

$$W_2(\mu_\tau, \nu_\tau) \le \exp\left(\int_0^\tau L\right) W_2(\mu_0, \nu_0) + \int_0^\tau \exp\left(\int_t^\tau L\right) \|u(t,\cdot;y) - v(t,\cdot;y)\|_{L^2(\nu_t)}\mathrm{d}t, \tag{20}$$

which is the endpoint-only distillation bound propagating the $t = 0$ mismatch to the whole trajectory.

**Corollary 9.** *(Few-step students: order-p one-step methods) Let the student be a one-step method (advancing to the next state using only the current state) of local order $p \ge 1$ with nodes $1 = t_1 \ge \cdots \ge t_{M+1} = 0$ and steps $h_k = t_k - t_{k+1}$. Assume that along student trajectories, the teacher drift $u$ has bounded time derivatives up to order $p$ and is $L(t)$-Lipschitz in $x$. Then there exist constants $\Gamma_k$ (depending on local $p$-th time derivatives of $u$ along teacher paths and spatial Lipschitz moduli $L(t)$) such that*

$$\int_{t_{k+1}}^{t_k} \|u(s,\cdot;y) - v(s,\cdot;y)\|_{L^2(\nu_s)}\mathrm{d}s \le \Gamma_k h_k^{p+1}. \tag{21}$$

*Consequently,*

$$W_2(\mu_\tau, \nu_\tau) \le \exp\left(\int_0^\tau L\right)\varepsilon + \exp\left(\int_0^\tau L\right)\sum_{k=1}^M \Gamma_k h_k^{p+1}, \tau \in [0, 1] \tag{22}$$

*For explicit Euler $(p = 1)$ and uniform steps $h_k = 1/M$, the second term scales like $O(M^{-1})$ up to smoothness constants.*

For explicit Euler $(p = 1)$, we take the student velocity $v(s,\cdot;y) = u(t_{k+1},\cdot;y)$ for all $s \in I_k = [t_{k+1}, t_k]$. Then, by the Fundamental Theorem of Calculus,

$$\|u(s,\cdot;y) - v(s,\cdot;y)\|_{L^2(\nu_s)} = \|u(s,\cdot;y) - u(t_{k+1},\cdot;y)\|_{L^2(\nu_s)} \tag{23}$$

$$\le \int_{t_{k+1}}^s \|\partial_t u(r,\cdot;y)\|_{L^2(\nu_r)}\mathrm{d}r. \tag{24}$$

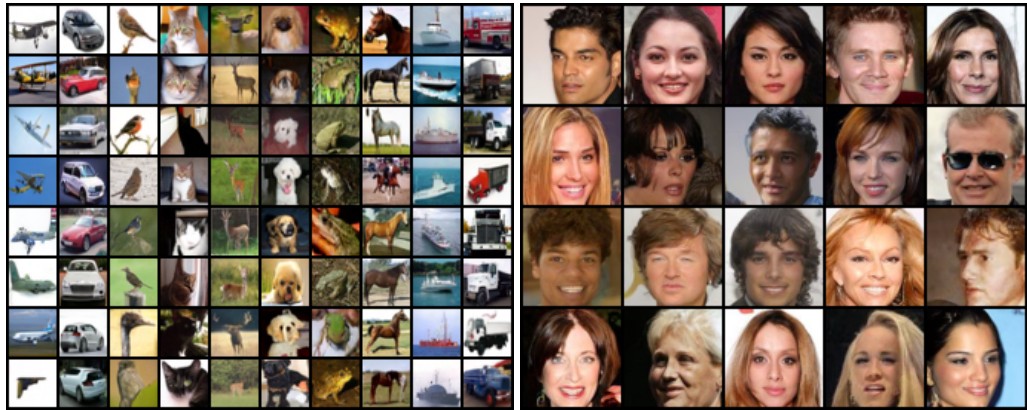

Figure 1: Uncurated CIFAR-10 (left) and CelebA-64 (right) samples generated by Guided Kac Flow (midpoint integrator, 100 steps, guidance $w = 3$ for CIFAR-10 and $w = 0$ for CelebA-64).

Integrating both sides over the interval $I_k$,

$$\int_{t_{k+1}}^{t_k} \|u - v\|_{L^2(\nu_s)} \mathrm{d}s \leq \int_{t_{k+1}}^{t_k} \int_{t_{k+1}}^{s} \|\partial_t u(r, \cdot; y)\|_{L^2(\nu_r)} \mathrm{d}r \mathrm{d}s \tag{25}$$

$$= \int_{t_{k+1}}^{t_k} \left( \int_{s=r}^{t_k} \mathrm{d}s \right) \|\partial_t u(r, \cdot; y)\|_{L^2(\nu_r)} \mathrm{d}r \quad \text{(by Tonelli's theorem)} \tag{26}$$

$$= \int_{t_{k+1}}^{t_k} (t_k - r) \|\partial_t u(r, \cdot; y)\|_{L^2(\nu_r)} \mathrm{d}r \tag{27}$$

$$\leq \left( \sup_{r \in I_k} \|\partial_t u(r, \cdot; y)\|_{L^2(\nu_r)} \right) \int_{t_{k+1}}^{t_k} (t_k - r) \mathrm{d}r \tag{28}$$

$$= \underbrace{\frac{1}{2} \left( \sup_{r \in I_k} \|\partial_t u(r, \cdot; y)\|_{L^2(\nu_r)} \right)}_{\Gamma_k} h_k^2 \tag{29}$$

we recover Equation (21).

For a first-order student, doubling the number of steps typically cuts the error by about half. For a higher-order student, the improvement is even faster. In practice, we balance efficiency and accuracy: higher-order students often require more evaluations per step, so we choose the order and step count to hit a target quality within a given compute budget. The constants depend on how smooth the teacher is and on the strength of guidance, but the qualitative rates are robust thanks to the finite-speed and bounded-energy structure of the Kac backbone.

## 5 EXPERIMENTS

### 5.1 SETUP

We train on CIFAR-10 (32×32), CelebA-64 (64×64), and LSUN Bedroom-256 (256×256). The velocity backbone is a UNet, and the hyperparameters and training details are provided in Appendix B.1. The procedure for selecting Kac flow hyperparameters is described in Appendix B.2. For classifier-free guidance, we use a constant guidance strength $w$ on CIFAR-10 and unconditional generation on CelebA-64 and LSUN Bedroom-256 (i.e., $w = 0$).

### 5.2 MAIN RESULTS

Our main CIFAR-10 results are summarized in Table 2. We train a conditional generative model and apply velocity guidance at evaluation, referred to as **Guided Kac Flow**. Figure 2 plots FID

| Method | CIFAR-10 | | | CelebA-64 | |
|---|---|---|---|---|---|
| | FID ↓ | NFE ↓ | Conditional | FID ↓ | NFE ↓ |
| Kac Flow (Duong et al., 2026) | 6.42 | 100 | | – | – |
| Guided Kac Flow (ours, 100 steps, midpoint) | 3.54 | 200 | ✓ | 3.36 | 200 |
| Guided Kac Flow (ours, 100 steps, AB-2) | 3.58 | 100 | ✓ | 3.50 | 100 |
| DistillKac (ours) | 3.72 | 20 | ✓ | 3.42 | 20 |
| | 4.14 | 4 | ✓ | 4.36 | 4 |
| | 4.68 | 2 | ✓ | 5.66 | 2 |
| | 5.66 | 1 | ✓ | 7.45 | 1 |
| DDPM (Ho et al., 2020) | 3.17 | 1000 | | – | – |
| DDIM (Song et al., 2021a) | 4.04 | 1000 | | 3.51 | 1000 |
| | 4.16 | 100 | | 6.53 | 100 |
| | 4.67 | 50 | | 9.17 | 50 |
| | 6.84 | 20 | | 13.73 | 20 |
| | 13.36 | 10 | | 17.33 | 10 |
| Progressive distillation (Salimans & Ho, 2022) | 2.57 | 8 | | – | – |
| | 3.00 | 4 | | – | – |
| | 4.51 | 2 | | – | – |
| | 9.12 | 1 | | – | – |
| EDM (Karras et al., 2022) | 1.79 | 35 | ✓ | – | – |
| | 1.97 | 35 | | – | – |
| iCT (Song & Dhariwal, 2024) | 2.46 | 2 | | – | – |
| | 2.83 | 1 | | – | – |
| Soft Diffusion (Daras et al., 2023) | – | – | | 1.85 | 300 |
| PDM-DDPM++ (Wang et al., 2023) | – | – | | 1.77 | 50 |
| Soft Truncation-G++ (Kim et al., 2023) | – | – | | 1.34 | 131 |

Table 2: **Main results on CIFAR-10 and CelebA-64.** Lower is better for FID. We report means over 50k samples. **Ours** denotes the Kac backbone with guidance and endpoint distillation. NFE counts velocity evaluations; midpoint integrator uses 2 function evaluations per step.

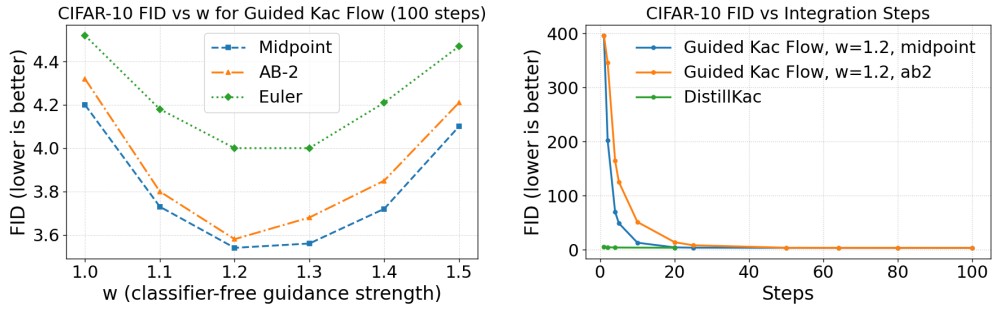

Figure 2: **Left:** CIFAR-10 FID vs. guidance strength $w$ for the 100-step Guided Kac Flow. Second-order integrators (midpoint and AB-2) outperform Euler. **Right:** CIFAR-10 FID vs. integration steps. DistillKac substantially reduces FID at 20, 4, 2, and 1 steps relative to Guided Kac Flow.

as a function of guidance strength $w$. We evaluate three time integrators: (1) Explicit Euler (first-order); (2) Midpoint/RK2 (second-order); (3) Adams-Bashforth-2 (AB-2; second-order). As shown in Figure 2, second-order methods, midpoint and AB-2, yield lower FID. Because AB-2 requires one function evaluation per step whereas midpoint requires two, AB-2 offers a better efficiency–accuracy trade-off.

Starting from a 100-step Guided Kac Flow model as a teacher, we distill a 20-step student and then iteratively distill to 4, 2, and 1 steps, each stage using the previously trained student as the teacher. We refer to these distilled students as **DistillKac**. Table 2 reports their FIDs. As shown in Figure 2,

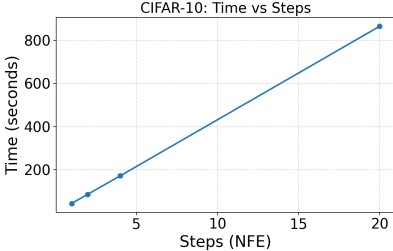 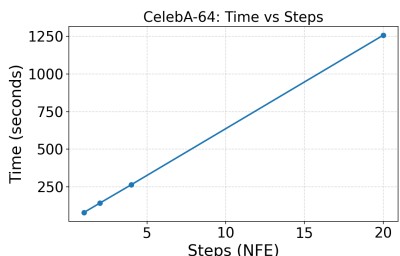

Figure 3: Time (seconds) to sample 50k images with one H100 GPU at 1, 2, 4, and 20 steps (NFE).

distillation significantly improves FID at 20, 4, 2, and 1 steps relative to the original Guided Kac Flow at the same step counts. DistillKac reduces the sampler from 100 to 1 step with FID rising only from 3.58 to 5.66 (+2.08). We attribute this robustness to the endpoint-to-trajectory stability of Kac flows (Theorem 8).

Analogously, we train an unconditional CelebA-64 Kac flow model. To distinguish it from DistillKac students, we still call it Guided Kac Flow teacher ($w = 0$). Starting from the teacher with 100 steps, we successively distill it to 20, 4, 2, and 1 steps. As the step count decreases, FID increases from 3.42 (20 NFE) to 7.45 (1 NFE). For comparison, the original Guided Kac Flow teacher yields 11.23 at 20 NFE and 443.01 at 1 NFE (See Appendix B.4). This indicates that distillation preserves image quality far better than the teacher in the few-step regime.

Multi-stage distillation ($S_1 \to S_2 \to \cdots \to S_\ell$) can perform better than a single-stage distillation ($S_1 \to S_\ell$) (See Appendix B.5). Multi-stage schedules (i) allow more hyperparameter tuning and (ii) reduce per-stage runtime, enabling more effective hyperparameter search.

## 6 RELATED WORK

Progressive Distillation (Salimans & Ho, 2022) iteratively halves the sampling steps by training a student to reproduce the effect of two teacher steps in one step. Consistency Models (Song et al., 2023) learn a single network with a multi-time consistency objective so that a sample can be produced in 1 or 2 evaluations via the probability flow ODE. Our endpoint-only distillation trains each student segment to match the teacher integrated over $N \geq 2$ substeps on that segment and provides an endpoint-to-trajectory stability guarantee under Kac dynamics. Conceptually, our scheme is closer to Progressive Distillation and, like Meng et al. (2023), we allow the teacher to employ classifier-free guidance.

Beyond pixel-space accelerated samplers, distillation of diffusion models in the latent space of a pre-trained autoencoder is widely used for large images. Meng et al. (2023) distills classifier-free guided diffusion models into few-step latent samplers. Latent Consistency Models (Luo et al., 2023) extend consistency models to the latent space of pre-trained latent diffusion models, such as Stable Diffusion. Latent Adversarial Diffusion Distillation (Sauer et al., 2024) distills a large latent diffusion transformer by training a student in latent space using the teacher both as a generator of synthetic guided samples and as a multi-scale, noise-aware discriminator backbone, enabling stable adversarial one-step or few-step sampling without decoding to pixels. Although we focus on pixel-space Kac backbones for image generation, our endpoint-distillation framework is representation-agnostic and could be applied equally well to latent Kac flows on top of a pre-trained VAE. In addition, our framework is architecture-agnostic, and can be applied not only to UNets but also to transformers and other architectures.

Our initial teacher, Guided Kac Flow, is a conditional Kac flow velocity model trained under hyperbolic, finite-speed Kac dynamics, i.e., a second-order system with an explicit velocity variable. By contrast, flow matching (Lipman et al., 2023) and rectified flow (Liu et al., 2023) learn first-order ODE velocity fields along chosen data–noise interpolants, with no independent velocity state or no wave-like propagation, and thus no built-in finite-speed constant. Moreover, classical diffusion has infinite propagation speed, and in learned diffusion models the score/velocity field becomes ill-

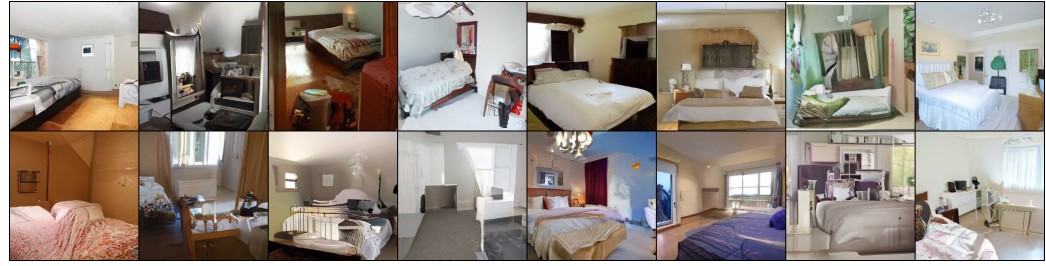

Figure 4: Uncurated LSUN Bedroom-256 samples generated by Guided Kac Flow (midpoint integrator, 100 steps, guidance $w = 0$).

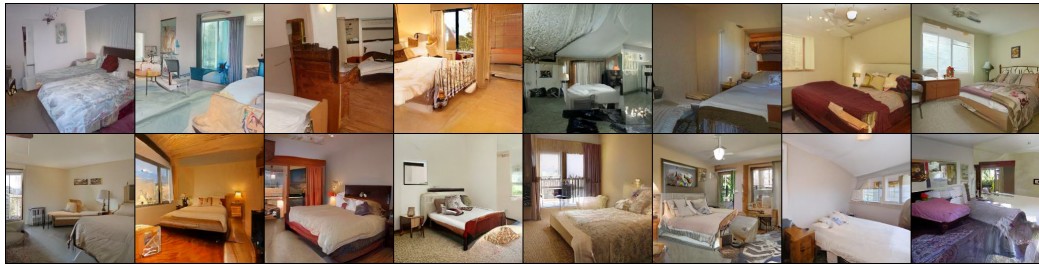

Figure 5: Uncurated LSUN Bedroom-256 samples generated by DistillKac (Euler integrator, 20 NFE).

conditioned as $\sigma \to 0$ (at the end of the denoising process), with Lipschitz blow-ups observed and analyzed (Yang et al., 2023; Duong et al., 2026).

## 7 DISCUSSION AND FUTURE WORK

While our CIFAR-10 and CelebA-64 FIDs are below the state-of-the-art diffusion models, we do not make SOTA claims. Instead, our results show that Guided Kac Flow and DistillKac produce competitive, high-fidelity images at low NFE. We expect further improvements from stronger backbones and additional tuning of the Kac flow hyperparameters. These are orthogonal to our core contribution. Our goal is to highlight Kac flow based generative modeling as a credible alternative to diffusion and to encourage follow-up work in this direction.

Unlike diffusion models, whose full probability path satisfies the Fokker-Planck equation, Kac flows enjoy a 1D Feynman-Kac correspondence with the telegrapher equation, yielding bounded-speed probability flow. As shown in Theorem 2, for higher dimensions ($d > 2$), the solution to the damped wave equation need not coincide with a probability density for the Kac process (and may fail to be a valid density). Consequently, the Kac flow paper (Duong et al., 2026) adopts a component-wise multi-D process built from independent 1D Kac marginals, for which the continuity equation and Lipschitz and finite-speed properties hold. This raises a natural open question: can one construct a genuinely $d$-dimensional stochastic process (with possibly dependent coordinates) whose probability path satisfies a hyperbolic PDE while preserving mass, perhaps under specific boundary conditions or coupling structures? Establishing such a model, along with stability guarantees and asymptotic connections to diffusion, would broaden the toolbox of finite-speed generative flows.

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

## LLM USAGE

**LLM use disclosure.** We used large language models to (i) improve the grammar and wording of this manuscript and (ii) assist with experimentation by drafting portions of the code and suggesting debugging steps. All scientific ideas, claims, experiment designs, and final implementations are by the authors. We reviewed and verified all LLM-assisted text and code before use, and the authors remain fully responsible for the content of this paper.

## A    EXTRA PROOFS

### A.1    PROOF OF THEOREM 4

By triangle inequality, for a.e. $t$,

$$\|\tilde{v}(t, \cdot; y)\|_{L^2(\mu_t)} \leq \|v_\theta(t, \cdot)\|_{L^2(\mu_t)} + |w(t)| \|\Delta_\theta(t, \cdot; y)\|_{L^2(\mu_t)} \tag{A-1}$$

$$\leq c\sqrt{d} + |w(t)| \|\Delta_\theta(t, \cdot; y)\|_{L^2(\mu_t)} \tag{A-2}$$

$$= c\sqrt{d} + |w(t)| \mathbb{E}_{x \sim \mu_t}[\|\Delta_\theta(t, x; y)\|^2] \tag{A-3}$$

$$< \infty \tag{A-4}$$

From Equation (A-1) to Equation (A-2), we used the assumption that $\|v_\theta(t, \cdot)\|_{L^2(\mu_t)} \leq c\sqrt{d}$ for a.e. $t$. From Equation (A-3) to Equation (A-4), we used Assumption 3 and the assumption that $|w(t)|$ is finite for a.e. $t$.

Therefore, $\tilde{v}(t, \cdot; y) \in L^2(\mu_t)$ for a.e. $t$.

### A.2    PROOF OF LEMMA 6

Let $x, z \in \mathbb{R}^d$ and $t \mapsto \Phi_{s \to t}(x), \Phi_{s \to t}(z)$ be teacher characteristics, i.e., the solution trajectories (integral curves) of the ODE defined by Equation (7). Let

$$\Delta(t) = \Phi_{s \to t}(x) - \Phi_{s \to t}(z). \tag{A-5}$$

Then

$$\frac{d}{dt}\Delta(t) = u(t, \Phi_{s \to t}(x); y) - u(t, \Phi_{s \to t}(z); y). \tag{A-6}$$

So

$$\frac{d}{dt}\|\Delta(t)\| = \left\langle \frac{\Delta(t)}{\|\Delta(t)\|}, \frac{d}{dt}\Delta(t) \right\rangle \tag{A-7}$$

$$\leq \left\| \frac{\Delta(t)}{\|\Delta(t)\|} \right\| \left\| \frac{d}{dt}\Delta(t) \right\| \quad \text{(by Cauchy–Schwarz inequality)} \tag{A-8}$$

$$= \|u(t, \Phi_{s \to t}(x); y) - u(t, \Phi_{s \to t}(z); y)\| \tag{A-9}$$

$$\leq L(t) \|\underbrace{\Phi_{s \to t}(x) - \Phi_{s \to t}(z)}_{\Delta(t)}\| \quad \text{(by Assumption 5)} \tag{A-10}$$

By Grönwall's inequality,

$$\|\Phi_{s \to t}(x) - \Phi_{s \to t}(z)\| \leq \exp\left(\int_s^t L(r)\mathrm{d}r\right) \|x - z\|, \tag{A-11}$$

hence

$$Lip(\Phi_{s \to t}) \leq \exp\left(\int_s^t L(r)\mathrm{d}r\right). \tag{A-12}$$

### A.3 PROOF OF LEMMA 7

Let $\pi$ be an optimal coupling between $\mu$ and $\nu$ for $W_2$. Then

$$W_2^2(F_\#\mu, F_\#\nu) \leq \int \|F(x) - F(z)\|_2^2 \mathrm{d}\pi(x,z) \quad \text{(by definition of } W_2\text{)} \tag{A-13}$$

$$\leq Lip(F)^2 \int \|x - z\|_2^2 \mathrm{d}\pi(x,z) \quad \text{(by definition of } Lip(F)\text{)} \tag{A-14}$$

$$= Lip(F)^2 W_2^2(\mu, \nu) \quad \text{(by definition of } W_2\text{)}. \tag{A-15}$$

Let $X \sim \nu$ and set $Y := F(X), Z := G(X)$. Let $\gamma$ be the joint distribution of $(Y, Z)$, i.e., $\gamma := (F \times G)_\#\nu$. Then $\gamma$ is a coupling of $F_\#\nu$ and $G_\#\nu$, i.e., $\gamma \in \Pi(F_\#\nu, G_\#\nu)$.

By the definition of pushforward,

$$\int_{\mathbb{R}^d \times \mathbb{R}^d} \|y - z\|^2 \mathrm{d}\gamma(y,z) = \int_{\mathbb{R}^d} \|F(x) - G(x)\|^2 \mathrm{d}\nu(x) = \mathbb{E}_{X\sim\nu}\|F(X) - G(X)\|^2 \tag{A-16}$$

By the definition of $W_2$,

$$W_2^2(F_\#\nu, G_\#\nu) = \inf_{\eta \in \Pi(F_\#\nu, G_\#\nu)} \int_{\mathbb{R}^d \times \mathbb{R}^d} \|y - z\|^2 \mathrm{d}\eta(y,z) \tag{A-17}$$

$$\leq \int_{\mathbb{R}^d \times \mathbb{R}^d} \|y - z\|^2 \mathrm{d}\gamma(y,z) \tag{A-18}$$

$$= \mathbb{E}_{X\sim\nu}\|F(X) - G(X)\|^2 \tag{A-19}$$

# B ADDITIONAL EXPERIMENTAL DETAILS

## B.1 EXPERIMENT CONFIGURATIONS

We use the UNet architecture from OpenAI's guided-diffusion github repository (OpenAI, 2021) to train models on CIFAR-10 and CelebA-64. Following Duong et al. (2026), we define a mean-reverting Kac process by

$$M_t := f(t)X_0 + K_{g(t)}, t \in [0, 1], \tag{B-1}$$

where $X_0 \sim \mu_0$ denotes the data, $K_t$ is a Kac process started from 0, and $f, g : [0, 1] \to \mathbb{R}$ are smooth functions satisfying

$$f(0) = 1, \quad f(1) = 0, \quad g(0) = 0, \quad g(1) = 1. \tag{B-2}$$

In particular, $M_0 = X_0$ and $M_1 = K_1$. Thus $f$ and $g$ are hyperparameters of the mean-reverting Kac process. Let $a := \frac{1}{2}\xi$, where $\xi$ is the damping coefficient in the damped wave equation (Equation (2)), and let $c$ denote the wave speed. The pair $(a, c)$ are additional hyperparameters of the mean-reverting Kac process. Table B-1 summarizes the UNet, training, and Kac process hyperparameters. We adopt early stopping when the validation FID curve flattens.

| Section | Parameter | CIFAR-10 | CelebA-64 | LSUN bedrooms-256 |
|---------|-----------|----------|-----------|-------------------|
| **UNet** | Input resolution | 32×32 | 64×64 | 256×256 |
| | Base channels | 256 | 192 | 256 |
| | Residual blocks / stage | 3 | 3 | 2 |
| | Dropout | 0.0 | 0.0 | 0.1 |
| | Head channels / attn | 64 | 64 | 64 |
| | Channel multipliers | [1, 2, 2, 3] | [1, 2, 3, 4] | [1, 1, 2, 2, 4, 4] |
| | Attention resolutions | [16, 8, 4] | [32, 16] | [32, 16, 8] |
| | Scale–shift norm | yes | yes | yes |
| | Resblock up/down | yes | yes | yes |
| | Use AMP | no | no | yes |
| **Training** | Optimizer | AdamW | AdamW | AdamW |
| | Learning rate | $10^{-4}$ | $2 \times 10^{-4}$ | $2 \times 10^{-4}$ |
| | Batch size | 128 | 128 | 64 |
| | Weight decay | 0.01 | 0.01 | 0.01 |
| | LR schedule | Flat $\to$ Cosine | Flat $\to$ Cosine | Flat $\to$ Cosine |
| | Warmup steps | 2% | 2% | 2% |
| | Flat steps | 58% | 58% | 58% |
| | Cosine steps | 40% | 40% | 40% |
| | Gradient clip | 1 | 1 | 1 |
| | EMA decay | 0.9999 | 0.9999 | 0.9999 |
| | Training steps | 300k–400k | 300k–400k | 300k–800k |
| | Early stopping | yes | yes | yes |
| **Kac** | $f(t)$ | $t$ | $t$ | $t$ |
| | $g(t)$ | $t$ | $t^2$ | $t^2$ |
| | $(a, c)$ | $(25, 2)$ | $(3000, 20)$ | $(3000, 20)$ |

Table B-1: UNet, training, and Kac process hyperparameters for CIFAR-10 and CelebA-64.

We visually inspect 100 generated samples every 5k training steps, and compute FID using 2,000 samples every 10k steps. For CIFAR-10, we trained on a single NVIDIA A100 or H100 GPU. The model typically converged after approximately 100k steps ($\approx$ 15 hours), at which point the training curve plateaued and we can apply early stopping. For CelebA-64, we trained on a single NVIDIA H100 GPU. The training curve plateaued after around 150k steps ($\approx$ 30 hours) and full training to 400k steps required approximately 80 hours. For LSUN Bedroom, we trained on 4×H100 GPUs. The training curve plateaued after about 300k steps ($\approx$ 3 days), but continued to improve very slowly thereafter. Extending training to 400k–800k steps required approximately 4–8 days in total.

### B.2 KAC FLOW HYPERPARAMETERS

As detailed in Appendix B.1, Kac flow hyperparameters comprise $a, c$ and the functions $f$ and $g$. We first use a lightweight UNet to search over different values of $a, c$, estimating FID with 2k samples to efficiently identify promising settings. We then adopt a deeper UNet described in B.1 and report final FID using 50k samples. Following Duong et al. (2026), we set $f(t) = t$, and consider $g(t) = t$ or $t^2$.

| $g(t) = t$ | | $g(t) = t^2$ | |
|---|---|---|---|
| $(a, c)$ | FID | $(a, c)$ | FID |
| $(25, 1)$ | 20.16 | $(900, 10)$ | 16.42 |
| $(25, 2)$ | 19.83 | $(1500, 10)$ | 16.60 |
| $(25, 3)$ | 19.38 | $(1500, 12)$ | 15.83 |
| $(25, 4)$ | 20.25 | $(3000, 10)$ | 18.11 |
| $(100, 5)$ | 36.23 | $(3000, 20)$ | **15.30** |
| $(900, 10)$ | 37.31 | $(6000, 20)$ | 16.96 |

Table B-2: FID@2k samples with a lightweight UNet on CelebA-64

Table B-2 reports FID computed from 2k samples using a lightweight UNet on CelebA-64. The set of hyperparameters $(a, c) = (3000, 20), f(t) = t, g(t) = t^2$ yields the lowest FID and can be adopted for subsequent experiments. The hyperparameters of the lightweight UNet are listed in Table B-3.

| | |
|---|---|
| Input resolution | 64×64 |
| Base channels | 128 |
| Residual blocks / stage | 2 |
| Dropout | 0.1 |
| Head channels / attn | 64 |
| Channel multipliers | [1, 2, 2, 2] |
| Attention resolutions | [16] |
| Scale–shift norm | no |
| Resblock up/down | yes |

Table B-3: Lightweight UNet Hyperparameters

### B.3 FID FOR GUIDED KAC FLOW ON CIFAR-10

Table B-4 and Table B-5 tabulate the FID values used to generate Figure 2. AB-2 uses one function evaluation per step whereas midpoint uses two evaluations per step.

| $w$ | 1.0 | 1.1 | 1.2 | 1.3 | 1.4 | 1.5 |
|---|---|---|---|---|---|---|
| midpoint | 4.20 | 3.73 | **3.54** | 3.56 | 3.72 | 4.10 |
| AB-2 | 4.32 | 3.80 | **3.58** | 3.68 | 3.85 | 4.21 |
| Euler | 4.52 | 4.18 | 4.00 | 4.00 | 4.21 | 4.47 |

Table B-4: FID for Guided Kac Flow (100 steps) on CIFAR-10

| steps | 100 | 80 | 64 | 50 | 25 | 20 | 10 | 5 | 4 | 2 | 1 |
|---|---|---|---|---|---|---|---|---|---|---|---|
| midpoint | 3.54 | 3.55 | 3.59 | 3.60 | 3.87 | 4.39 | 13.03 | 49.14 | 70.36 | 202.37 | 396.46 |
| AB-2 | 3.58 | 3.62 | 3.67 | 3.77 | 8.25 | 13.99 | 51.36 | 125.63 | 165.16 | 346.16 | 396.38 |

Table B-5: FID vs. integration steps for Guided Kac Flow ($w = 1.2$) on CIFAR-10.

### B.4    FID for Guided Kac Flow on CelebA-64

Table B-6 reports FID for the Guided Kac Flow teacher ($w = 0$) across various NFE using the midpoint and AB-2 integrators. AB-2 uses one function evaluation per step, whereas midpoint uses two evaluations per step. Consequently, midpoint cannot be run at NFE = 1. Figure B-1 shows FID vs. NFE for both the Guided Kac Flow teacher ($w = 0$) and the DistillKac student. DistillKac substantially reduces FID at 20, 4, 2, and 1 steps relative to the Guided Kac Flow teacher ($w = 0$).

| NFE | 100 | 20 | 4 | 2 | 1 |
|---|---|---|---|---|---|
| midpoint | 3.61 | 11.31 | 71.28 | 443.01 | |
| AB-2 | 3.50 | 11.23 | 56.73 | 190.03 | 443.01 |

Table B-6: FID for the Guided Kac Flow teacher ($w = 0$) on CelebA-64.

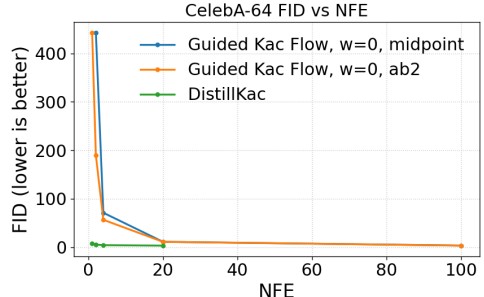

Figure B-1: CelebA-64 FID vs. NFE. DistillKac substantially reduces FID at 20, 4, 2, and 1 steps relative to the Guided Kac Flow teacher ($w = 0$).

### B.5    Multi-stage Distillation vs Single-stage Distillation

Table B-7 shows that multi-stage distillation ($S_1 \to S_2 \to \cdots \to S_\ell$) performs better than a single-stage distillation ($S_1 \to S_\ell$), where $S_i$ is the step count of the model. In Table B-7, $4 \to 2 \to 1$ means we start from a 4-step DistillKac and successively distill it to 2 and 1 steps, while $4 \to 1$ means we directly distill the same 4-step DistillKac to 1 step. Similarly, $100 \to 20 \to 4 \to 2 \to 1$ means we start from a Guided Kac Flow teacher with 100 steps and progressively distill it to 20, 4, 2, and 1 steps, while $100 \to 1$ means we directly distill the same Guided Kac Flow teacher to 1 step. At the distillation stage $S_i \to S_{i+1}$, the teacher substep is the quotient $S_i/S_{i+1}$ (assuming $S_{i+1}|S_i$), i.e., for each sample, the student advances one step while the teacher must simulate $S_i/S_{i+1}$ steps. The runtime at this stage is proportional to $S_i/S_{i+1}$. Therefore, multi-stage distillation has a shorter per-stage runtime and makes hyperparameter search more tractable.

| $w$ | 1.0 | 1.1 | 1.2 | 1.3 | 1.4 | 1.5 |
|---|---|---|---|---|---|---|
| $4 \to 2 \to 1$ | 6.95 | 5.78 | **5.66** | 6.01 | 6.57 | 7.22 |
| $4 \to 1$ | 6.71 | 5.74 | 5.72 | 6.13 | 6.77 | 7.40 |
| $100 \to 20 \to 4 \to 2 \to 1$ | 6.95 | 5.78 | **5.66** | 6.01 | 6.57 | 7.22 |
| $100 \to 1$ | 13.06 | 10.83 | 9.71 | 9.34 | 9.40 | 9.67 |

Table B-7: FID for multi-stage distillation and single-stage distillation.

### B.6    Empirical Evidence for Endpoint-to-trajectory Stability

We compute the mean squared discrepancy between the Guided Kac Flow teacher ($w = 0$) and DistillKac students with $M \in \{20, 4, 2, 1\}$ steps (See Figure B-2). For each DistillKac student with $M$ steps, we start from the same initial noise and run both the teacher and the student. We then

evaluate the teacher at the $M$ student time steps $\{t_k\}_{k=1}^M$ and compute the mean, over $n$ samples, of $\|x_k - \hat{x}_k\|^2$ across these time steps, where $x_k$ and $\hat{x}_k$ denote the teacher and student states at time step $t_k$. As shown in Figure B-2, the mean squared teacher–student discrepancy grows as $t$ approaches 0, and the 20-step DistillKac student is consistently closer to the teacher than the 4-, 2-, and 1-step DistillKac students. This is consistent with the endpoint-to-trajectory stability bound in Theorem 8: once the student matches the teacher at segment endpoints, the trajectory mismatch remains controlled throughout the interval.

To further quantify this effect, we plot, over many random initial noises, the mean trajectory discrepancy versus the endpoint discrepancy (See Figure B-3). For each sample, the mean trajectory discrepancy is computed as the average of $\|x_k - \hat{x}_k\|$ over all teacher time steps $\{t_k\}_{k=1}^K$, and the endpoint discrepancy is $\|x_K - \hat{x}_K\|$ at the final time step $t_K$. The scattered points cluster tightly around a straight line, indicating an approximately linear relationship between trajectory and endpoint errors. Empirically, endpoint mismatch almost completely explains the observed trajectory discrepancy, supporting our endpoint-only distillation scheme.

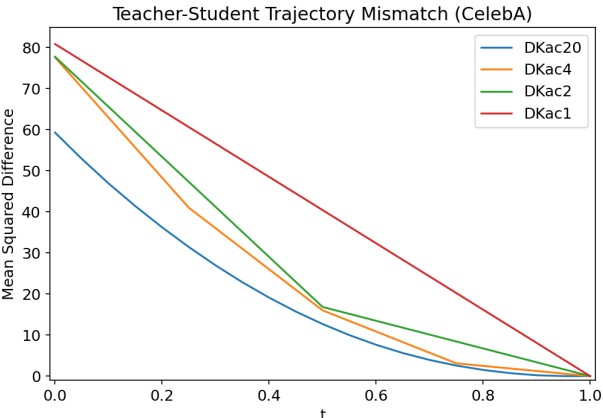

Figure B-2: Mean squared trajectory mismatch between the Guided Kac Flow teacher ($w = 0$) and DistillKac students with 20, 4, 2, and 1 steps. Here, DKac20, DKac4, DKac2, and DKac1 denote DistillKac students with 20, 4, 2, and 1 steps, respectively.

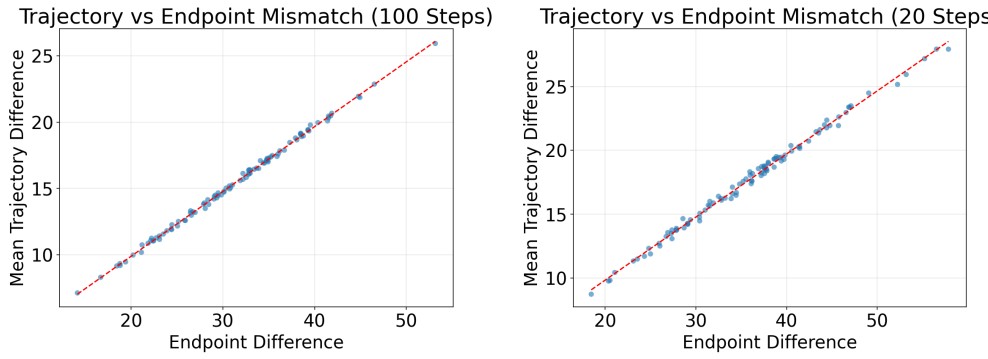

Figure B-3: Mean trajectory discrepancy versus endpoint discrepancy over 100 samples. Each blue dot corresponds to one sample, and we fit a straight line to the scattered points. **Left:** Comparison between the Guided Kac Flow teacher ($w = 0$) and the DistillKac student with 20 steps. The mean trajectory discrepancy is computed over 100 teacher steps. The points have Pearson correlation 0.9992 and linear regression $R^2 = 0.9983$. **Right:** Comparison between DistillKac students with 20 and 4 steps. The mean trajectory discrepancy is computed over 20 teacher steps. The points have Pearson correlation 0.9976 and linear regression $R^2 = 0.9952$.

## B.7 MORE GENERATED IMAGES

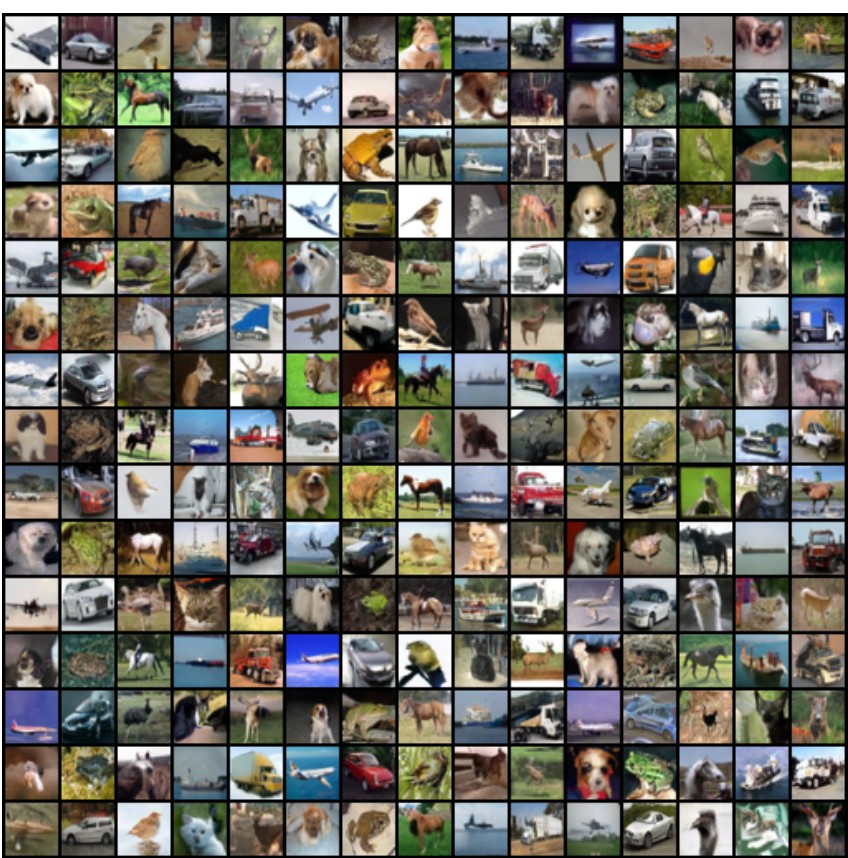

Figure B-4: Uncurated CIFAR-10 samples generated by DistillKac (Euler integrator, 20 NFE).

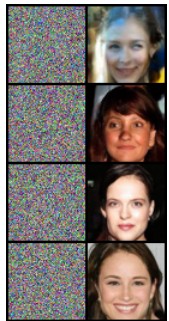 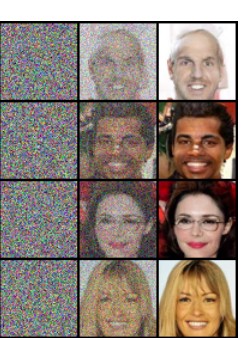 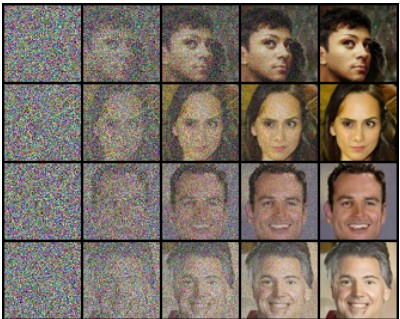

Figure B-5: Uncurated CelebA-64 generations with 1, 2, and 4 steps (NFE).

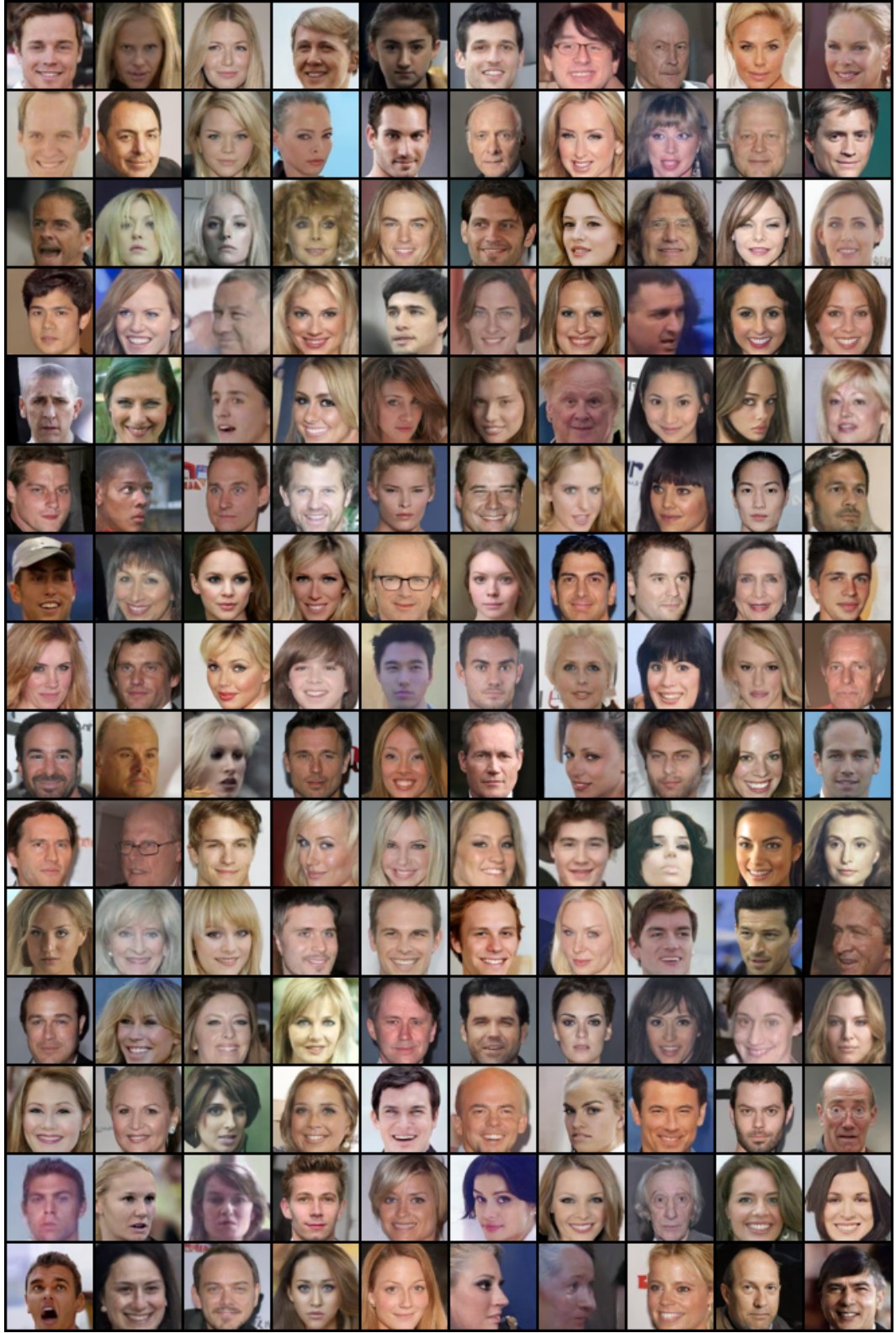

Figure B-6: Uncurated CelebA-64 samples generated by DistillKac (AB-2 integrator, 20 NFE).

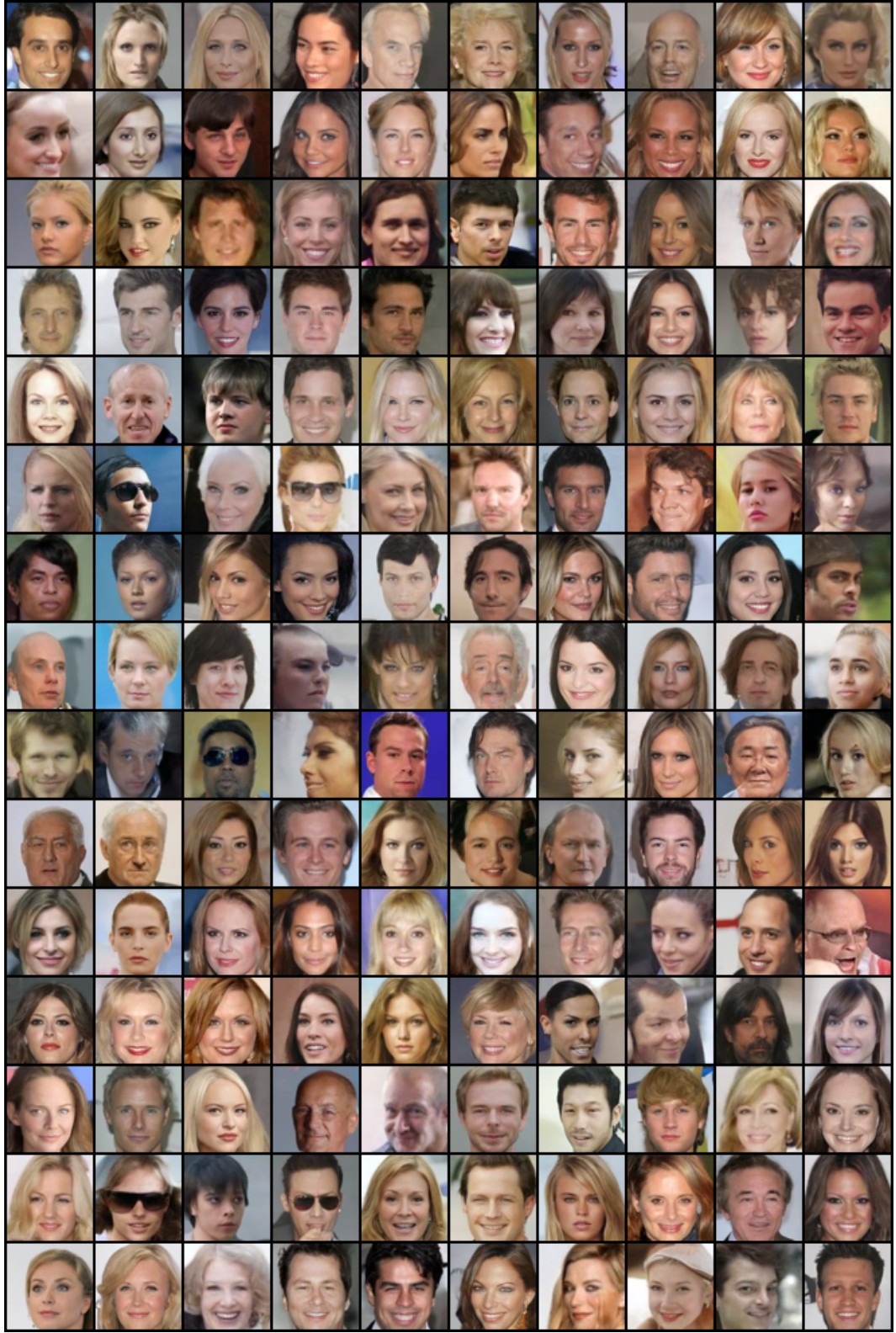

Figure B-7: Uncurated CelebA-64 samples generated by DistillKac (Euler integrator, 4 NFE).

