# OpenReview forum: "DistillKac: Few-Step Image Generation via Damped Wave Equations"
_ICLR.cc/2026/Conference — ICLR 2026 Poster_

### Official Review · Reviewer_eWqj · 2025-10-28

**Soundness:** 3
**Presentation:** 3
**Contribution:** 2
**Rating:** 2
**Confidence:** 4

**Summary:**

The authors propose DistillKac a strategy for distillation for generative models whose densities evolve with the damped wave equation. This parameterization should remove the singularities at the end points which is common in diffusion/flow models. This achieved by fixing the propagation speed to some bounded value. The authors then propose a distillation strategy for this existing family of models.

**Strengths:**

I liked the detailed maths and found the Kac representation to be interesting.

* Theorem 8 and Corollary 9 seem interesting tools for analysis in the literature on (consistency) distillation.
* Theorem 4 seems novel within the context of CFG
* Addressing the problems of singularities in diffusion/flow models is important

**Weaknesses:**

## Primary concerns
### Theory
* How does this work clearly differentiate from [1]? It is unclear where [1] ends and where the contributions of this work begin. My rough understanding is that this work is [1] plus Theorems 4, 8 and 9 with distillation?
* I'm surprised that learning 1-D independent coordinate processes yields anything useful. It seems strange that this works as there is often interactions between the different coordinates. This seems to be an **incredibly strong** assumption to make.
* How do you initially train the teacher? Flow matching and diffusion models are nice because of simulation-free training, it is not clear to me that the Kac representation yields this for the teacher.
* Also since in Section 2.2 the process $X(t)$ is defined as a jump process it doesn't seem right to use standard numerical schemes for ODEs (like Euler or Heun) to integrate teacher model as suggested in line 203.
* Also since the Kac process is stochastic why are the flow maps governed by an ODE, shouldn't the flow maps be random variables *a la* [3]?

### Empirical
* Both CIFAR-10 and CelebA-64 are quite small and generally considered toy datasets, in particular since one of the contributions is improved CFG I feel like more compelling experiments for guidance should be included than just CIFAR-10 (32 x 32).
* The empirical results don't look great for Kac Flows. The traditional diffusion models seem to still outperform them despite the singularity issues, so what is the advantage? This seems especially true when compared to consistency models.
* Overall, the empirical results are wholly **uncompelling** in convincing the reader of the advantages of Kac flows over standard flow/diffusion models.

## Minor comments / suggestions
* Table 1 should get reworked it is unclear what each row corresponds to. Some labels would be nice, e.g., PDE, SDE, papers.
* Since there is a mixture of prior results in the theoretical sections, maybe highlight your contributions with a colored box?
* FID is a notoriously fickle and unwieldy metric to work with. Recent papers looking at text-guided generation [2] have included CLIP-score  [4] and Image Reward [5].

## References
[1] Richard Duong, Jannis Chemseddine, Peter K Friz, and Gabriele Steidl. Telegrapher’s generative
model via kac flows. arXiv preprint arXiv:2506.20641v3, 2025.

[2] Skreta et al., 2025, Feynman-Kac Correctors in Diffusion:
Annealing, Guidance, and Product of Experts, ICML, https://arxiv.org/pdf/2503.02819

[3] Kunita, Hiroshi. Stochastic flows and jump-diffusions. Springer Singapore, 2019.

[4] Radford, Alec, et al. "Learning transferable visual models from natural language supervision." International conference on machine learning. PmLR, 2021.

[5] Xu, Jiazheng, et al. "Imagereward: Learning and evaluating human preferences for text-to-image generation." Advances in Neural Information Processing Systems 36 (2023): 15903-15935.

**Questions:**

1. Why do you refer to the random paths and evolution of the densities as the *Kac process* and *Kac flow* (resp.)? On line 117 you say when $d \geq 2$ such processes are called *random flights*. Aren't you working with $d \geq 2$ objects? The terminology is a bit confusing
2. How does this work relate to [1]?
3. Do Theorem 8 and Corollary 9 apply to any flow maps or only those admitted by the *Kac process*?
4. Likewise, how does Theorem 8 compare to [2, Theorem 4.1] outside of W1 vs W2?
5. How do you justify the coordinate-wise independence?
6. Please address any questions raised in the weaknesses sections.


[1] Skreta et al., 2025, Feynman-Kac Correctors in Diffusion:
Annealing, Guidance, and Product of Experts, ICML, https://arxiv.org/pdf/2503.02819

[2] Dou et al., 2024, Provable Statistical Rates for Consistency Diffusion Models, https://arxiv.org/pdf/2406.16213

---

> ### Public Comment · ~Richard_Duong1 · 2025-11-22
>
> As one of the authors of the referenced paper [1], I would like to comment on the review of Reviewer eWqj, and clarify some of the misunderstandings of this review concerning the Kac process.
>
> > Also since in Section 2.2 the process $X(t)$ is defined as a jump process it doesn't seem right to use standard numerical schemes for ODEs (like Euler or Heun) to integrate teacher model as suggested in line 203.
>
> Neither the Kac process considered in [1], nor the process $X(t)$ considered in Section 2.2 of the Submission13819, are jump processes. They are stochastic processes with a.s. continuous paths; their *time derivatives* are jump processes. Hence, there are no additional problems using standard numerical schemes for ODEs (like Euler).
>
> > I'm surprised that learning 1-D independent coordinate processes yields anything useful. It seems strange that this works as there is often interactions between the different coordinates. This seems to be an incredibly strong assumption to make.
>
> The independence assumption on the coordinate processes refers to the *latent variables*, and is no different to the usual approaches of Flow Matching/Diffusion models which rely on isotropic Gaussian latents. On the data coordinates, there are *no* assumptions on independence in [1] (and also in Submission13819 as far as I can tell).
>
> > Also since the Kac process is stochastic why are the flow maps governed by an ODE, shouldn't the flow maps be random variables a la [3]?
>
> This question, closely related to the reviewer's Question 1, reflects a misunderstanding of standard concepts in generative flow modeling. The Kac process is a stochastic process, which (like any other process) induces an evolution of probability distributions/densities $p_t$, which themselves are *not random* anymore. This evolution $(p_t)$, also adressed as "probability flow", gives rise to a so-called "flow ODE" and a flow map, which are deterministic and not random. Only the function composition of the flow map and latent variable renders another random variable.
>
> Given these conceptual issues, the current Confidence rating 4 may not accurately reflect the reviewer’s familiarity with this area.

---

> > ### Author Response · Authors · 2025-12-03
> >
> > We greatly appreciate the author of the referenced paper [1] for taking the time to clarify several technical misunderstandings raised in the review and for their supportive comments on our work.
> >
> > [1] Richard Duong, Jannis Chemseddine, Peter K Friz, and Gabriele Steidl. Telegrapher’s generative model via kac flows. arXiv preprint arXiv:2506.20641v3, 2025.

---

> ### Comment · Reviewer_eWqj · 2025-11-23
>
> I appreciate the commenter's clarifications.
>
> > Neither the Kac process considered in [1], nor the process considered in Section 2.2 of the Submission13819, are jump processes. They are stochastic processes with a.s. continuous paths; their time derivatives are jump processes. Hence, there are no additional problems using standard numerical schemes for ODEs (like Euler).
>
> I reread Section 2.2 and yes the commenter is right, it is the time derivative of $X(t)$ that is a jump process not $X(t)$ itself, but I still question the application of ODE integrators to this stochastic process.
>
> > The independence assumption on the coordinate processes refers to the latent variables, and is no different to the usual approaches of Flow Matching/Diffusion models which rely on isotropic Gaussian latents. On the data coordinates, there are no assumptions on independence in [1] (and also in Submission13819 as far as I can tell).
>
> I reread Section 2.3 and from the best I can tell from this section each component of $\boldsymbol X = (X_1, X_2, \ldots, X_d)$ evolves independently from time $T$ to time $0$. *I.e.*, $X_i^{s,t,x} = \boldsymbol \Phi_{s \to t}^{(i)}(x_i)$ for $i$-th coordinate and initial value $x$. Whilst, I know that in flow/diffusion models we often assume the source distribution is an isotropic Gaussian distribution, the assertion in this paper seems much stronger. For a flow model an equivalent assertion seems to be that we can decompose our vector field $\boldsymbol u_\theta(\boldsymbol x)$ into $\boldsymbol u_\theta = (u_1, u_2, \ldots, u_d)$ which each only take $x_i$ as in input. This would mean that $\boldsymbol u$ has a diagonal Jacobian, this does seem like a rather strong assertion. If the authors only meant that the source distribution has this coordinate-wise independence I think it could be more clearly stated in the writing of Section 2.3.
>
> >  This question, closely related to the reviewer's Question 1, reflects a misunderstanding of standard concepts in generative flow modeling. The Kac process is a stochastic process, which (like any other process) induces an evolution of probability distributions/densities , which themselves are not random anymore. This evolution , also adressed as "probability flow", gives rise to a so-called "flow ODE" and a flow map, which are deterministic and not random. Only the function composition of the flow map and latent variable renders another random variable.
>
> Yes I understand that just like the Fokker-Planck-Kolmogorov equations there is a tie between the deterministic evolution of the probability densities $p_t$ and the random process $\boldsymbol X_t$ via the Feynman-Kac formula. I am also aware of Song and Maoutsa's result for the probability flow ODE which describes an ODE evolution of which has the same marginals $p_t$ as the reverse-time score SDE. So it depends for diffusion models, are we using the ODE or SDE formulation? If the ODE formulation then what the commenter wrote is true, otherwise for the SDE case we can view the flow maps $\\{\boldsymbol \Phi_{s,t}\\}$ as a stochastic flow of diffeomorphisms [1]. From Equation (3) in the paper I see that $\boldsymbol X(t)$ evolves with a stochastic process because $U(t)$ is a jump process. Thus a deterministic flow map seems in inappropriate when there is a stochastic component.
>
> [1] Li, Xuechen, et al. "Scalable gradients for stochastic differential equations." International Conference on Artificial Intelligence and Statistics. PMLR, 2020.

---

> > ### Author Response · Authors · 2025-12-03
> > **Author Response (4/N)**
> >
> > > I reread Section 2.2 and yes the commenter is right, it is the time derivative of  that is a jump process not  itself, but I still question the application of ODE integrators to this stochastic process.
> >
> > Our paper uses two different objects: (1) The stochastic Kac process defined in Section 2.2 and (2) The deterministic probability flow ODE whose flow we actually integrate at inference. We never apply ODE integrators to the stochastic process. Instead, in training, we use the Kac process only through its analytic transition density and conditional velocity, and in inference (teacher and student), generation is performed by solving the deterministic probability flow ODE backwards in time, exactly in the same sense as “probability flow ODE” for diffusion models. ODE integrators are applied only to this ODE.
> >
> > > I reread Section 2.3 and from the best I can tell from this section [...]
> >
> > Only the latent Kac noise process has independent coordinates. The data is arbitrary and can have rich dependencies across coordinates. The interpolated process inherits those dependencies from data and hence its coordinates are not independent unless the data itself is.
> >
> > This is conceptually the same as diffusion / flow matching using an isotropic Gaussian or standard Brownian motion as the reference, where the latent noise has independent coordinates, but the learned flow and the induced distributions may be highly dependent.
> >
> > The component‑wise construction for the reference process does not impose any factorization on the learned velocity field or on the probability flow ODE. So this does not force a diagonal Jacobian for the learned vector field.
> >
> > > Yes I understand that just like the Fokker-Planck-Kolmogorov equations [...]
> >
> > Here again, the key distinction is which evolution we associate a flow map to. Equation (3) defines the stochastic Kac process. In Sections 4 we then explicitly introduce deterministic flow maps as the solution operators of the ODEs (7)-(8). These are the standard deterministic flows associated with the probability flow ODE, completely analogous to the ODE flows used for diffusion (Song et al., 2020).
> >
> > **Reference**
> > - Song, Yang, Jascha Sohl-Dickstein, Diederik P. Kingma, Abhishek Kumar, Stefano Ermon, and Ben Poole. "Score-based generative modeling through stochastic differential equations." arXiv preprint arXiv:2011.13456 (2020).

---

> ### Author Response · Authors · 2025-12-03
> **Author Response (1/N)**
>
> > How does this work clearly differentiate from [1]? It is unclear where [1] ends and where the contributions of this work begin. My rough understanding is that this work is [1] plus Theorems 4, 8 and 9 with distillation?
>
> Duong et al. (2025) introduce Kac flows as a new backbone for generative modeling, providing an analytic conditional velocity for 1‑D Kac processes which is then extended to images via independent 1‑D components. They train unconditional Kac flows via conditional flow matching and compare against diffusion baselines.
>
> Our work builds on top of that backbone in two orthogonal directions:
> 1. Guided Kac Flows: We adapt classifier‑free guidance to the Kac setting directly in velocity space. We show that as long as the base Kac velocity has finite kinetic energy and $w(t)$ is finite a.e., the guided velocity still has finite kinetic energy. Conceptually: we provide a theoretical guarantee that classifier‑free guidance does not create the kinetic‑energy blow‑ups seen in diffusion flows near data time, leveraging the finite‑speed hyperbolic backbone. This guided‑Kac analysis is not present in Duong et al.
> 2. Endpoint‑only distillation with trajectory stability. We propose DistillKac, which distills a Kac teacher (possibly guided) into a student by endpoint matching only. We then prove Theorem 8 (endpoint‑to‑trajectory stability), which shows that under a mild spatial Lipschitz assumption on the teacher and student drifts (Assumption 5), a small discrepancy at some time plus an integral of drift differences controls the Wasserstein‑2 distance between teacher and student for all later times. Corollary 9 further specializes this to explicit Euler and higher‑order integrators.
>
> Empirically, we have a systematic study of guided Kac flows and DistillKac students on CIFAR‑10, CelebA‑64, and LSUN Bedroom‑256, including comparisons with diffusion and consistency‑style baselines and ablations supporting the theory (e.g., the near‑linear relationship between endpoint discrepancy and mean trajectory discrepancy in Figure B‑3).
>
> These results are not in Duong et al. (2025), which analyzes the Kac backbone itself but neither considers classifier‑free guidance nor endpoint‑distilled few‑step students, nor proves endpoint‑to‑trajectory stability for distilled flows.
>
> > I'm surprised that learning 1-D independent coordinate processes yields anything useful. It seems strange that this works as there is often interactions between the different coordinates. This seems to be an incredibly strong assumption to make.
>
> The independence assumption is only on the reference noise process, not on the data distribution or the learned model. Our multi‑dimensional Kac process is built as a product of 1‑D Kac processes along each coordinate, following Duong et al. (2025). The data distribution is not factorized; it can exhibit arbitrary dependencies across coordinates/pixels. The velocity network is a UNet acting on the full image, and there is no architectural factorization across coordinates. Cross‑coordinate interactions are learned by the network just as in diffusion models, where the forward Brownian noise is also independent per coordinate.
>
> In other words, the assumption is comparable to diffusion models using independent Brownian noise per pixel. It simplifies the analytic PDE/probability‑flow connection and yields explicit finite‑speed and energy bounds, but the learned generative model is not coordinate‑factorized.
>
> > How do you initially train the teacher? Flow matching and diffusion models are nice because of simulation-free training, it is not clear to me that the Kac representation yields this for the teacher.
>
> Training follows the conditional flow matching recipe (Duong et al. 2025; Lipman et al., 2022).
> This is analogous in spirit to diffusion/flow matching: we do not simulate sample paths of the Kac process numerically. We only need its one‑time marginals and conditional velocities, which are available in closed form.
>
>
> **Reference**
> - Richard Duong, Jannis Chemseddine, Peter K Friz, and Gabriele Steidl. Telegrapher’s generative model via kac flows. arXiv preprint arXiv:2506.20641v3, 2025.
>
> - Lipman, Yaron, Ricky TQ Chen, Heli Ben-Hamu, Maximilian Nickel, and Matt Le. "Flow matching for generative modeling." arXiv preprint arXiv:2210.02747 (2022).

---

> ### Author Response · Authors · 2025-12-03
> **Author Response (2/N)**
>
> > Also since in Section 2.2 the process $X(t)$ is defined as a jump process it doesn't seem right to use standard numerical schemes for ODEs (like Euler or Heun) to integrate teacher model as suggested in line 203.
>
> Duong et al. prove that the Kac process induces an absolutely continuous curve in Wasserstein space satisfying a continuity equation with bounded velocity field. This is directly analogous to the probability flow ODE in diffusion models (Song et al., 2020), where a forward SDE and a deterministic ODE share the same marginal path.
>
> Our generator is a probability flow ODE and we integrate this deterministic ODE using standard numerical schemes (Euler, midpoint, AB‑2). We are not numerically simulating the underlying stochastic process. We are solving the deterministic probability flow ODE, for which ODE solvers are exactly appropriate.
>
> > Also since the Kac process is stochastic why are the flow maps governed by an ODE, shouldn't the flow maps be random variables a la [3]? [3] Kunita, Hiroshi. Stochastic flows and jump-diffusions. Springer Singapore, 2019.
>
> Kunita’s stochastic flow theory considers random diffeomorphisms generated by SDEs and jump‑diffusions. Our analysis, however, is in the standard deterministic ODE setting: once the drift is fixed, the flow maps are deterministic maps, and the randomness comes only from the initial condition.
>
> This is exactly the framework used for probability flow ODEs in diffusion and flow matching: the randomness is in the initial sample, not in the flow map. Our Theorem 8 therefore uses standard ODE stability tools rather than stochastic flow theory.
>
> > Do Theorem 8 and Corollary 9 apply to any flow maps or only those admitted by the Kac process?
>
> Theorem 8 and Corollary 9 are stated and proved for general flows whose drifts satisfy Assumption 5. No Kac‑specific structure is used in the proofs. Kac flows only enter when we want concrete dimension‑aware bounds on $L(t)$ and the kinetic energy. Thus, Theorem 8 and Corollary 9 apply to any probability flow ODE, including diffusion probability flows or other flow‑matching backbones, that satisfies the same Lipschitz and regularity conditions.
>
> > Likewise, how does Theorem 8 compare to [2, Theorem 4.1] outside of W1 vs W2? [2] Dou et al., 2024, Provable Statistical Rates for Consistency Diffusion Models, https://arxiv.org/pdf/2406.16213
>
> Theorem 4.1 of Dou et al. is basically saying: If you train a consistency model by distilling from a diffusion model, and your assumptions (nice tails, Lipschitz score, etc.) hold, then the distance between the learned model’s distribution and the true data distribution (measured in Wasserstein-1) can be tightly bounded, and this bound is essentially controlled by how well you estimated the score function of the original diffusion model.
>
> In contrast, our Theorem 8 analyzes how closely a few‑step student tracks a fixed teacher flow, assuming the teacher itself is already well‑trained.
>
> > How does this work relate to [1]? [1] Skreta et al., 2025, Feynman-Kac Correctors in Diffusion: Annealing, Guidance, and Product of Experts, ICML, https://arxiv.org/pdf/2503.02819
>
> [1] introduces Feynman–Kac Correctors (FKCs), a principled way to modify diffusion model sampling so that heuristic score combinations (like classifier-free guidance, annealing, or product-of-experts) actually follow the desired intermediate target distributions, by augmenting the SDE with importance weights and Sequential Monte Carlo resampling. Using this framework, the authors show improved text-to-image guidance, temperature-annealed sampling from Boltzmann densities, and multi-objective molecule generation without retraining the underlying diffusion models.
>
> So [1] is quite different from our work.
>
> > How do you justify the coordinate-wise independence?
>
> This is answered above.
>
>
> **Reference**
> - Richard Duong, Jannis Chemseddine, Peter K Friz, and Gabriele Steidl. Telegrapher’s generative model via kac flows. arXiv preprint arXiv:2506.20641v3, 2025.
>
> - Song, Yang, Jascha Sohl-Dickstein, Diederik P. Kingma, Abhishek Kumar, Stefano Ermon, and Ben Poole. "Score-based generative modeling through stochastic differential equations." arXiv preprint arXiv:2011.13456 (2020).

---

> ### Author Response · Authors · 2025-12-03
> **Author Response (3/N)**
>
> > Why do you refer to the random paths and evolution of the densities as the Kac process and Kac flow (resp.)? On line 117 you say when $d \geq 2$ such processes are called random flights. Aren't you working with $d \geq 2$ objects? The terminology is a bit confusing
>
> In our original submission, Section 2.2 states that in 1D the process is called the telegraph or Kac process, while in higher dimensions it is called a random flight. In Section 2.4 we then define our terminology: “We use Kac process when we talk about individual random paths (simulation/noising), and Kac flow when we talk about how distributions move in time and the ODE we integrate at inference.”
>
> > Both CIFAR-10 and CelebA-64 are quite small and generally considered toy datasets, in particular since one of the contributions is improved CFG I feel like more compelling experiments for guidance should be included than just CIFAR-10 (32 x 32).
>
> We have expanded the experiments in the revision to include LSUN Bedroom‑256 (256×256). For classifier‑free guidance, we focused on CIFAR‑10 because: (i) it is the standard benchmark for conditional image generation, and (ii) it allows systematic sweeps over guidance strength and integrators under a tight compute budget.
>
> > The empirical results don't look great for Kac Flows. The traditional diffusion models seem to still outperform them despite the singularity issues, so what is the advantage? This seems especially true when compared to consistency models.
>
> As stated in our original submission, our goal is not to claim state‑of‑the‑art FID. The results in our manuscript should be read as evidence that the finite‑speed Kac backbone can support robust few‑step samplers with strong sample quality, not as an attempt to beat every diffusion or flow model.
>
> > FID is a notoriously fickle and unwieldy metric to work with. Recent papers looking at text-guided generation [2] have included CLIP-score [4] and Image Reward [5].
>
> The metrics you reference (CLIP‑score, ImageReward) are primarily designed for text‑guided generation. Our experiments are class‑conditional (CIFAR‑10) and unconditional (CelebA‑64, LSUN Bedroom-256), where FID remains the standard metric used in prior Kac work and almost all comparable baselines we cite.
>
> > Since there is a mixture of prior results in the theoretical sections, maybe highlight your contributions with a colored box?
>
> In our original submission, our main contributions are stated at the end of Section 1.

---

### Official Review · Reviewer_FCnC · 2025-10-29

**Soundness:** 3
**Presentation:** 3
**Contribution:** 3
**Rating:** 4
**Confidence:** 4

**Summary:**

DistillKac presents an image generative model grounded in the damped wave equation, a hyperbolic PDE represented stochastically through the Kac process. Unlike diffusion-based ODEs or SDEs, whose velocity norms can diverge near the terminal time, Kac dynamics maintain globally bounded kinetic energy and Lipschitz continuity in Wasserstein space. The main contributions include adapting classifier-free guidance directly in velocity space while preserving bounded-energy guarantees, and introducing an endpoint-only distillation scheme with a theoretical stability bound linking endpoint consistency to full-trajectory alignment. Empirically, these features enable stable few-step sampling with competitive image quality, and theoretically, the work clarifies why endpoint matching is sufficient under finite-speed Kac dynamics.

**Strengths:**

- The paper is clearly written and presents solid theoretical foundations with rigorous stability proofs.
- Adopting a hyperbolic PDE instead of the conventional diffusion formulation offers a novel structural perspective in generative modeling, characterized by finite propagation speed and bounded velocity.
- The introduction of Kac flow–based generative modeling provides an innovative alternative to diffusion approaches, and the inclusion of stability guarantees and asymptotic links to diffusion broadens the methodological toolkit for finite-speed generative flows.

**Weaknesses:**

- While this paper provides solid theoretical formulations and formal stability proofs, it lacks empirical analysis demonstrating how Kac flow concretely improves stability in practice. Theoretical claims of bounded energy and finite propagation speed are not supported by detailed quantitative studies.
- Although Table 1 attempts to ensure fairness by matching the number of function evaluations (NFEs) across methods, the comparison would be more convincing if it also included training time, FLOPs, or wall-clock cost, since these reflect actual computational efficiency.
- The paper does not show whether the claimed stability advantages translate into observable robustness improvements, such as reduced divergence, fewer NaNs, or more consistent quality under large classifier-free guidance scales or few-step sampling.

**Questions:**

Overall, the work would benefit from comprehensive runtime and stability analyses to substantiate that the proposed Kac flow offers practical, measurable improvements beyond theoretical guarantees.

---

> ### Author Response · Authors · 2025-12-03
>
> We thank the reviewer for the feedback.
>
> > While this paper provides solid theoretical formulations and formal stability proofs, it lacks empirical analysis demonstrating how Kac flow concretely improves stability in practice. [...]
>
> We added a new section in the appendix (Appendix B.6) titled “Empirical evidence for endpoint-to-trajectory stability.” There we provide two quantitative experiments on CelebA-64:
> 1. Teacher–student trajectory discrepancy vs. time: For DistillKac students with 20, 4, 2, and 1 steps, we start from the same noise, run both the 100-step Guided Kac Flow teacher and the student, evaluate the teacher on the student time grid, and compute the mean squared discrepancy as a function of time. We observe that: (1) The discrepancy grows gradually as $t \to 0$ (toward data time), and (2) The 20-step student is consistently closer to the teacher than the 4-, 2- and 1-step students.
> 2. Mean trajectory discrepancy vs. endpoint discrepancy: For many samples, we compute the mean trajectory error over all teacher steps and the endpoint error at the final time. The scatter plots lie very close to a straight line. In other words, endpoint discrepancy almost perfectly predicts trajectory discrepancy.
>
> These measurements directly support the theoretical claim that controlling endpoints suffices to control the whole path in our setting. Together with the FID vs. step-count comparisons, these diagnostics show that endpoint-only distillation under Kac dynamics yields a much milder quality degradation when collapsing from 100 to 1-4 steps than naive step reduction.
>
> > Although Table 1 attempts to ensure fairness by matching the number of function evaluations (NFEs) across methods, the comparison would be more convincing if it also included training time, FLOPs, or wall-clock cost, since these reflect actual computational efficiency.
>
> Do you mean “Table 2” instead of “Table 1”? In our original submission, we have already plotted time (seconds) to sample 50k images at 1, 2, 4, and 20 steps (NFE) in Figure 3. Additionally, in our original submission, we have already included training steps in Appendix B1. To further address your concern, in Appendix B1 of our revised manuscript, we added the wall-clock training budgets for CIFAR-10, CelebA-64, and LSUN Bedroom-256.
>
> > The paper does not show whether the claimed stability advantages translate into observable robustness improvements, such as reduced divergence, fewer NaNs, or more consistent quality under large classifier-free guidance scales or few-step sampling.
>
> Our experiments already test robustness along several axes.
> 1. Behaviour under classifier‑free guidance. Figure 2 (left) and Table B‑4 show FID as a function of classifier‑free guidance strength $w$ for the 100‑step Guided Kac Flow on CIFAR‑10. Across $w \in [1.0, 1.5]$ all integrators remain numerically stable. Second‑order methods (midpoint and AB‑2) achieve the best FID, but there is no evidence of divergence or collapse as $w$ increases within this range. This empirical behaviour is consistent with Theorem 4, which guarantees finite kinetic energy for any finite $w$.
> 2. Few‑step robustness. DistillKac is explicitly aimed at the very low‑NFE regime, which is typically where numerical issues manifest. As discussed above, FID degrades gracefully from 3.58 (100 steps) to 5.66 (1 step) on CIFAR‑10 and from 3.50 (100 NFE) to 7.45 (1 NFE) on CelebA‑64, while the undistilled teacher collapses to FID > 400 at 1 step. This demonstrates that our distilled Kac flows retain sample quality and do not exhibit numerical blow‑ups even with extremely aggressive step reduction.
> 3. Trajectory‑level diagnostics. The new Appendix B.6 experiment is directly targeted at the reviewer’s question about “observable robustness.” It measures how the teacher–student discrepancy evolves over time and shows (i) smooth growth along the trajectory and (ii) an almost linear relationship between endpoint and trajectory error (Fig. B‑2, B‑3), which strongly supports the practical relevance of our stability theory.

---

### Official Review · Reviewer_H4Mg · 2025-11-04

**Soundness:** 3
**Presentation:** 3
**Contribution:** 3
**Rating:** 6
**Confidence:** 3

**Summary:**

The paper presents a fast image generator, DistillKac,  that uses the damped wave equation and its stochastic Kac representation to move probability mass at finite speed. Based on that, it introduces classifier-free guidance in velocity space that preserves square integrability and proposes endpoint-only distillation that trains a student to match a frozen teacher over long intervals.

**Strengths:**

1. Explore a new equation form (damped wave equation) for the diffusion process.
2. Provide a solid and detailed proof of the formulation, including the error bounds.

**Weaknesses:**

1. If I’m getting this right, the comparison in Figure 2 is made against normal sampling, specifically the non-distilled version of the model at the same number of steps. But doesn’t that seem a bit unfair? Plus, similar distillation methods can also achieve just a slight increase in FID. How does that show that your distillation method is better? It might be helpful to include comparisons with other distillation methods.

2. In the related work section, you only mention a few papers that focus on CIFAR-10 and CelebA-64, but there’s actually a lot of literature on distillation in latent space that has algorithms similar to yours. For instance, the skipping steps in LCM[1] have the same meaning as N in your Algorithm 1. It would be great to see some mention and analysis of those similarities.

3. While the proof goes into detail about the error and stability bounds, the experiments don’t seem to show better error and stability compared to other distillation methods. This leaves us unsure about the advantages of your proposed distillation method in practice.

[1]Luo, Simian, et al. "Latent consistency models: Synthesizing high-resolution images with few-step inference." arXiv preprint arXiv:2310.04378 (2023).

**Questions:**

see weakness.

---

> ### Author Response · Authors · 2025-12-03
>
> We thank the reviewer for the detailed and constructive feedback. We address the weaknesses and questions point by point below.
>
> > If I’m getting this right, the comparison in Figure 2 is made against normal sampling, specifically the non-distilled version of the model at the same number of steps. But doesn’t that seem a bit unfair? [...]
>
> Our goal in Figure 2 (Right) is not to claim that DistillKac strictly dominates all existing distillation methods. We are simply answering this question: “If I want few-step sampling with a Kac backbone, how much does distillation help compared to simply reducing the number of steps of the original teacher model?”
>
> Regarding comparisons with other distillation schemes, Table 2 already includes representative diffusion-based accelerators, such as Progressive Distillation, EDM, and iCT on CIFAR-10.
>
> > [...] there’s actually a lot of literature on distillation in latent space that has algorithms similar to yours [...]
>
> We agree and have substantially expanded the related work to cover latent-space distillation and to highlight the connection you mention.
>
> In Related Work Section, we now explicitly discuss:
> - Meng et al. (2023), which distills classifier-free guided diffusion models into few-step latent samplers.
> - Latent Consistency Models (LCM; Luo et al., 2023), which bring consistency training into the latent space of pre-trained latent diffusion models, such as Stable Diffusion.
> - Latent Adversarial Diffusion Distillation (LADD; Sauer et al., 2024), which distills a latent diffusion transformer by using the teacher both as a generator of synthetic guided samples and as a multi-scale, noise-aware discriminator backbone, enabling stable one- or few-step latent sampling.
>
> At the same time, we highlight that our framework is representation-agnostic: while the experiments use pixel-space Kac flows, the theory and Algorithm 1 only assume a Kac velocity field and do not depend on whether this is in pixel or latent space. We now state explicitly that the same endpoint-distillation scheme could be applied to latent Kac flows on top of a pre-trained VAE.
>
> > [...] the experiments don’t seem to show better error and stability compared to other distillation methods [...]
>
> We added a new section in the appendix (Appendix B.6) titled “Empirical evidence for endpoint-to-trajectory stability.” There we provide two quantitative experiments on CelebA-64:
>
> 1. Teacher–student trajectory discrepancy vs. time: For DistillKac students with 20, 4, 2, and 1 steps, we start from the same noise, run both the 100-step Guided Kac Flow teacher and the student, evaluate the teacher on the student time grid, and compute the mean squared discrepancy as a function of time. We observe that:
> (1) The discrepancy grows gradually as $t \to 0$ (toward data time), and
> (2) The 20-step student is consistently closer to the teacher than the 4-, 2- and 1-step students.
>
> 2. Mean trajectory discrepancy vs. endpoint discrepancy: For many samples, we compute the mean trajectory error over all teacher steps and the endpoint error at the final time. The scatter plots lie very close to a straight line. In other words, endpoint discrepancy almost perfectly predicts trajectory discrepancy.
>
> These measurements directly support the theoretical claim that controlling endpoints suffices to control the whole path in our setting. Together with the FID vs. step-count comparisons, these diagnostics show that endpoint-only distillation under Kac dynamics yields a much milder quality degradation when collapsing from 100 to 1-4 steps than naive step reduction.
>
> We agree that a full head-to-head comparison of such stability diagnostics across all existing distillation schemes would be valuable, but it would require re-implementing those methods under the Kac backbone, which is beyond the scope of this work. Our aim here is to show that the specific structure of finite-speed Kac flows leads to a distillation procedure whose empirical behavior closely matches its theoretical stability guarantees.

---

### Official Review · Reviewer_wRPC · 2025-11-07

**Soundness:** 3
**Presentation:** 3
**Contribution:** 3
**Rating:** 6
**Confidence:** 3

**Summary:**

This paper introduces DistillKac, a new framework for fast image generation based on damped wave equations and their stochastic Kac representation. Unlike diffusion models that rely on parabolic Fokker–Planck dynamics, DistillKac models generative flows as finite-speed hyperbolic PDEs, ensuring bounded kinetic energy and improved numerical stability. The authors extend this formulation with (1) velocity-space classifier-free guidance, which allows controlled generation without destabilizing the dynamics, and (2) endpoint distillation, a novel few-step training strategy where a student model learns from the teacher’s terminal distribution rather than stepwise trajectories.

**Strengths:**

1. The theoretical formulation is elegant and original, bridging kinetic transport theory (via the Kac process) with modern generative modeling. Modeling probability flow through damped wave equations offers a compelling alternative to diffusion-based parabolic PDEs.
2. The proposed endpoint distillation is conceptually simple yet powerful, backed by a provable endpoint-to-trajectory stability guarantee. This provides a rare theoretical justification for few-step model distillation.
3. The velocity-space classifier-free guidance is well-motivated and resolves instability issues common in diffusion guidance, maintaining bounded energy across time.

**Weaknesses:**

1. While mathematically grounded, the physical intuition behind using wave dynamics for generative flow could be developed further — especially how finite-speed propagation impacts sample diversity or convergence in practice.
2. The evaluation scope is modest, focusing mainly on CIFAR-10 and CelebA-64. Demonstrations on higher-resolution or more complex datasets (e.g., LSUN, ImageNet) would strengthen the empirical case.
3. The architecture choice (UNet) and the fixed set of integration schemes limit understanding of how well DistillKac generalizes to larger backbones (e.g., DiT) or different ODE solvers.

**Questions:**

1. Can the authors provide more empirical evidence for the endpoint-to-trajectory stability theorem, e.g., quantitative correlation between endpoint and full-trajectory discrepancies?
2. How does finite propagation speed affect perceptual smoothness or texture diversity compared to diffusion’s infinite-speed propagation?

---

> ### Author Response · Authors · 2025-12-03
>
> We thank the reviewer for the detailed and constructive feedback. We address the weaknesses and questions point by point below.
>
> > While mathematically grounded, the physical intuition behind using wave dynamics for generative flow could be developed further [...]
>
> Intuitively, Kac / telegrapher dynamics enforce a finite propagation speed: starting from localized mass, probability can only spread inside a “causal cone,” and the induced velocity field has globally bounded kinetic energy. In practice, this means that along the generative trajectory, each step can only move the state by at most a controlled amount; the flow cannot suddenly “jump” or become arbitrarily stiff near data time. This contrasts with diffusion / Fokker-Planck dynamics, where mass spreads instantaneously and the reverse-time drift is known to become very stiff near terminal time.
>
> > The evaluation scope is modest, focusing mainly on CIFAR-10 and CelebA-64. [...]
>
> While our theory, including the endpoint-to-trajectory stability theorem, holds regardless of the dataset, we agree that testing on more complex datasets would further strengthen the empirical case. In the revised manuscript we therefore added LSUN Bedroom-256 (256x256) experiments. We train the same Kac backbone on LSUN Bedrooms and report qualitative results at 256x256, including uncurated grids for both the 100-step Guided Kac Flow teacher and few-step DistillKac students. We keep the architecture as a UNet, with appropriate channel multipliers for 256x256, but otherwise do not introduce LSUN-specific tricks.
>
> > The architecture choice (UNet) and the fixed set of integration schemes limit understanding of how well DistillKac generalizes to larger backbones (e.g., DiT) or different ODE solvers.
>
> Our experiments use a UNet backbone and a set of explicit ODE solvers (Euler, midpoint/RK2, Adams–Bashforth-2) to keep the comparison focused, but the theory and algorithm are not tied to this specific architecture.
> 1. Architecture: In the revised manuscript we explicitly state that the endpoint-distillation framework is architecture-agnostic. The analysis is done at the level of the continuous-time velocity field and only assumes Lipschitz/smoothness properties. Any backbone capable of parameterizing such a velocity field (e.g., DiT-style transformers or larger UNets, in pixel or latent space) can in principle be used. We also note that the framework is representation-agnostic, and could be applied on top of a pre-trained VAE in latent space, similar to existing latent distillation works.
> 2. ODE solvers: The endpoint-to-trajectory stability theorem itself is solver-agnostic and applies to any reasonable integration method.
>
> > Can the authors provide more empirical evidence for the endpoint-to-trajectory stability theorem [...]
>
> We added a new section in the appendix (Appendix B.6) titled “Empirical evidence for endpoint-to-trajectory stability.” There we provide two quantitative experiments on CelebA-64:
>
> 1. Teacher–student trajectory discrepancy vs. time:
> For DistillKac students with 20, 4, 2, and 1 steps, we start from the same noise, run both the 100-step Guided Kac Flow teacher and the student, evaluate the teacher on the student time grid, and compute the mean squared discrepancy as a function of time. We observe that:
> (1) The discrepancy grows gradually as $t \to 0$ (toward data time), and
> (2) The 20-step student is consistently closer to the teacher than the 4-, 2- and 1-step students.
>
> 2. Mean trajectory discrepancy vs. endpoint discrepancy:
> For many samples, we compute the mean trajectory error over all teacher steps and the endpoint error at the final time. The scatter plots lie very close to a straight line. In other words, endpoint discrepancy almost perfectly predicts trajectory discrepancy.
>
> These measurements directly support the theoretical claim that controlling endpoints suffices to control the whole path in our setting.
>
> > How does finite propagation speed affect perceptual smoothness or texture diversity [...]
>
> Our main evidence is indirect: competitive FIDs versus diffusion-based few-step methods on CIFAR-10 and CelebA-64, and visually diverse uncurated samples (varied objects, faces, and room layouts). We do not claim that finite-speed dynamics automatically outperform diffusion in perceptual smoothness or diversity. Rather, the finite-speed structure makes it possible to reduce NFE very aggressively without inducing mode collapse or severe texture artifacts.

---

### Author Response · Authors · 2025-12-03

We thank the reviewers for their constructive feedback. As explained in the original paper, we stress that our goal is not to claim SOTA FID, but to demonstrate that hyperbolic, finite-speed Kac flows plus endpoint distillation yield competitive, stable few-step samplers, and open up a complementary PDE perspective to diffusion. The stability theorem for our endpoint-only distillation (Theorem 8) is derived under conditions that are naturally satisfied by finite-speed Kac flows. Thanks to their bounded energy and Lipschitz regularity, Kac dynamics admit an endpoint-to-trajectory stability guarantee for distillation. To the best of our knowledge, an analogous finite-speed, endpoint-based stability result has not been established for standard diffusion backbones, whose reverse-time dynamics typically lack these properties.

In response to reviews, we have revised the manuscript along three main axes:

**1. Expanded related work and clearer positioning.**

We now explicitly connect DistillKac to the latent-space distillation literature, and clarify that our endpoint-distillation scheme is both representation-agnostic (pixel or latent space) and architecture-agnostic (UNet, transformers, etc.).

**2. New experiments and stronger empirical evidence.**

**Higher-resolution images.** We add LSUN Bedroom-256 (256x256) experiments, including qualitative samples for both the Guided Kac Flow teacher and DistillKac students, demonstrating that our Kac backbone and endpoint distillation remain stable and produce coherent 256x256 images at low NFE.

**Improved CelebA-64 few-step performance.** With additional tuning of Kac hyperparameters and the distillation schedule, we obtain better FID for 2- and 1-step students on CelebA-64 compared to the original submission (Table 2), while keeping the same teacher.

**Empirical endpoint-to-trajectory stability.** To support Theorem 8, we add experiments measuring teacher-student mismatch along the whole path versus at the endpoint. The new figures show that (i) trajectory error grows in a controlled way as $t \to 0$ and (ii) mean trajectory discrepancy is almost linearly determined by endpoint error, with near-perfect correlation, providing empirical evidence for endpoint-only supervision.

**3. More detailed training budgets.**

We clarify the wall-clock training budgets for CIFAR-10, CelebA-64, and LSUN Bedroom-256.

---

### Meta-Review · Area_Chair_QPWd · 2026-01-05

**Summary:**

This paper focuses on the few-step generative model with Kac representation. The proposed method, DistillKac, uses the damped wave equation and its stochastic Kac representation to move probability mass at finite speed. Based on the representation, this paper introduces   classifier free guidance in velocity space and endpoint only distillation to achieve better performance. The paper is rated as 6,6,4, and 2 intially, with main concerns focusing on theoretical disagreement and lack of enough experimental demonstration. During rebuttal, the authors and external commentor clarify the misunderstanding of the theoretical disagreement. In addition, more experiments and comparison demonstrate the effectiveness of the proposed method. AC confirms that the concerns are mostly addressed and recommends this paper to **Accept (Poster)**.

**Reviewer Concerns:**

In the initial review, main concerns focus on theoretical disagreement and lack of enough experimental demonstration.

During rebuttal, authors provide additional results on higher-resolution images (LSUN) and quantitative results on endpoint-to-trajectory stability. The concerns on the experimental demonstration are mostly addressed.

Reviewer eWqj raises concerns about Kac process. Authors and external commentor (first author of the main referenced paper) clarify the misunderstanding. Reviewer eWqj still not quite agrees but AC confirms that the concern is mostly addressed.

**Reviewer Scores:**

**Reviewer wRPC** keeps 6. Reviewer wRPC did not have chance to reply to the authors, but AC confirms that the concerns are mostly addressed.

**Reviewer H4Mg** keeps 6. Reviewer H4Mg did not have chance to reply to the authors, but AC confirms that the concerns are mostly addressed.

**Reviewer FCnC** raises to 6. Reviewer FCnC did not have chance to reply to the authors, but AC confirms that the concerns are mostly addressed.

**Reviewer eWqj** raises to 4. The main concerns of Reviewer eWqj lie in two aspects: theoretical and empirical. There still be disagreement on the theoretical part but the empirical part is mostly addressed.

---

### Decision · Program_Chairs · 2026-01-26

Accept (Poster)